# Dietary Olive Leaf Extract Differentially Modulates Antioxidant Defense of Normal and *Aeromonas hydrophila*-Infected Common Carp (*Cyprinus carpio*) via Keap1/Nrf2 Pathway Signaling: A Phytochemical and Biological Link

**DOI:** 10.3390/ani13132229

**Published:** 2023-07-06

**Authors:** Doaa H. Assar, Amany E. Ragab, Essam Abdelsatar, Abdallah S. Salah, Shimaa M. R. Salem, Basma M. Hendam, Soad Al Jaouni, Rasha A. Al Wakeel, Marwa F. AbdEl-Kader, Zizy I. Elbialy

**Affiliations:** 1Clinical Pathology Department, Faculty of Veterinary Medicine, Kafrelsheikh University, Kafrelsheikh 33516, Egypt; 2Pharmacognosy Department, Faculty of Pharmacy, Tanta University, Tanta 32527, Egypt; amany.ragab@pharm.tanta.edu.eg; 3Pharmacognosy Department, Faculty of Pharmacy, Cairo University, Cairo 11562, Egypt; essam.abdelsattar@pharma.cu.edu.eg; 4Department of Aquaculture, Faculty of Aquatic and Fisheries Sciences, Kafrelsheikh University, Kafrelsheikh 33516, Egypt; abdallah_salah_2014@fsh.kfs.edu.eg; 5Institute of Aquaculture, Faculty of Natural Sciences, University of Stirling, Stirling FK9 4LA, UK; 6Department of Animal Nutrition and Nutritional Deficiency Diseases, Faculty of Veterinary Medicine, Mansoura University, Mansoura 35516, Egypt; shimaaradi_2009@mans.edu.eg; 7Department of Animal Wealth Development, Faculty of Veterinary Medicine, Mansoura University, Mansoura 35516, Egypt; basmahendam@mans.edu.eg; 8Department of Hematology/Pediatric Oncology, Yousef Abdulatif Jameel Scientific Chair of Prophetic Medicine Application, Faculty of Medicine, King Abdulaziz University, Jeddah 21589, Saudi Arabia; saljaouni@kau.edu.sa; 9Department of Physiology, Faculty of Veterinary Medicine, Kafrelsheikh University, Kafrelsheikh 33516, Egypt; rasha_elwakel2010@vet.kfs.edu.eg; 10Department of Fish Health and Management, Sakha Aquaculture Research Unit, Central Laboratory for Aquaculture Research, A.R.C., Kafrelsheikh 33516, Egypt; marwa.abdelkader@vet.kfs.edu.eg; 11Department of Fish Processing and Biotechnology, Faculty of Aquatic and Fisheries Sciences, Kafrelsheikh University, Kafrelsheikh 33516, Egypt

**Keywords:** olive leaf extract, *C. carpio*, *Aeromonas hydrophila*, antioxidant, immunostimulant, gene expression, UPLC-PDA-MS/MS

## Abstract

**Simple Summary:**

*A. hydrophila* is a pathogenic agent not only for fish species but for mammals, including humans, especially those in contact with sick fish as well. To combat this infectious pathogen, the OLE was used to study its effect on the growth performance, immunohematological, antioxidant, and histopathological changes in *C. carpio* both under normal conditions and after the *A. hydrophila* infection. Our study indicated that OLE, when used in the *C. carpio* diets at a level of 0.1%, has a high capacity for improving growth performance by boosting nutrient utilization, increasing the antioxidant system activity in the cells and improving the immune response. However, higher doses of the OLE must be avoided since they may induce oxidative stress. Therefore, the OLE at a dose of 0.1% is considered an organic natural product without adverse effects on the environment or the fish as a safe and sustainable alternative to chemical growth promoters in aquaculture practice.

**Abstract:**

Olive leaves are an immense source of antioxidant and antimicrobial bioactive constituents. This study investigated the effects of dietary incorporation of olive leaf extract (OLE) on the growth performance, hematobiochemical parameters, immune response, antioxidant defense, histopathological changes, and some growth- and immune-related genes in the common carp (*Cyprinus carpio*). A total of 180 fish were allocated into four groups with triplicate each. The control group received the basal diet without OLE, while the other three groups were fed a basal diet with the OLE at 0.1, 0.2, and 0.3%, respectively. The feeding study lasted for 8 weeks, then fish were challenged with *Aeromonas hydrophila*. The results revealed that the group supplied with the 0.1% OLE significantly exhibited a higher final body weight (FBW), weight gain (WG%), and specific growth rate (SGR) with a decreased feed conversion ratio (FCR) compared to the other groups (*p* < 0.05). An increase in immune response was also observed in the fish from this group, with higher lysosome activity, immunoglobulin (IgM), and respiratory burst than nonsupplemented fish, both before and after the *A. hydrophila* challenge (*p* < 0.05). Similarly, the supplementation of the 0.1% OLE also promoted the *C. carpio’s* digestive capacity pre- and post-challenge, presenting the highest activity of protease and alkaline phosphatase (*p* < 0.05). In addition, this dose of the OLE enhanced fish antioxidant capacity through an increase in the activity of superoxide dismutase (SOD) and glutathione peroxidase (GPx) and decreased hepatic lipid peroxidation end products (malondialdehyde—MDA), when compared to the control group, both pre- and post-infection (*p* < 0.05). Concomitantly with the superior immune response and antioxidant capacity, the fish fed the 0.1% OLE revealed the highest survival rate after the challenge with *A. hydrophila* (*p* < 0.05). A significant remarkable upregulation of the hepatic *sod*, *nrf2*, and protein kinase C transcription levels was detected as a vital approach for the prevention of both oxidative stress and inflammation compared to the infected unsupplied control group (*p* < 0.05). Interestingly, HPLC and UPLC-ESI-MS/MS analyses recognized that oleuropein is the main constituent (20.4%) with other 45 compounds in addition to tentative identification of two new compounds, namely oleuroside-10-carboxylic acid (I) and demethyl oleuroside-10-carboxylic acid (II). These constituents may be responsible for the OLE exerted potential effects. To conclude, the OLE at a dose range of 0.66–0.83 g/kg *w*/*w* can be included in the *C. carpio* diet to improve the growth, antioxidant capacity, and immune response under normal health conditions along with regulating the infection-associated pro-inflammatory gene expressions, thus enhancing resistance against *A. hydrophila.*

## 1. Introduction

To meet the demand for dietary protein, aquaculture is considered one of the food production sectors with the quickest economic growth [1,2]. However, zoonotic risks and numerous infectious diseases that put this business in danger have been issues for aquaculture [3,4]. Chemotherapeutic medication has been utilized in aquaculture as a convenient and simple method. Despite its advantages, overusing antibiotics in aquaculture systems can have detrimental effects on both health and the economy [5] leading to the appearance of many antibiotic-resistant bacteria and drug-resistant genes in the aquatic environment of both fish and shellfish [6,7]. To meet the human food consumption, we have to sustainably boost growth rates in aquaculture, replace or reduce the use of antibiotics, prevent and control fish diseases, and improve the environmental protection to be safe for both human and animal health [8,9,10,11]. Immunostimulant agents are one of these substitutes that help aquatic animals to grow and survive by preventing opportunistic diseases through enhancing their natural defense and antioxidant processes [10,12,13]. 

Metabolites that have been found in plants are known as phytochemicals. Phytochemicals are classified, based on their chemical structure, into different classes of which flavonoids, phenolics, and terpenoids are biologically important [14]. Phytochemicals may exert an antioxidant effect by providing electrons to prevent other molecules from being oxidized [14]. A natural antioxidant defense system present inside the body comprises diverse antioxidant enzymes such as superoxide dismutase, glutathione reductase, catalase, and glucose-6-phosphate dehydrogenase as a complex immune system defends against oxidative damage [15,16]. By neutralizing free radicals, these endogenous antioxidants sustain a cellular redox state under typical circumstances. The ability of endogenous cellular antioxidant defenses can be exceeded, though, if endogenous antioxidants are insufficient as a result of exposure to stimuli that enhance the formation of oxidants, such as the emergence of chronic diseases, pollution, injury, or exercise [17]. In such cases, the body has to be replenished with exogenous phytochemicals or other antioxidants. For instance, maintaining cellular redox status and balancing the oxidation of proteins, lipids, and DNA may acquire the intake of phenolic antioxidants, carotenoids, minerals, and vitamins [18,19]. Due to their many benefits, including decreased environmental dangers, lack of drug resistance, low cost, and availability, medicinal plants are favored as natural alternatives [13,20,21,22,23].

The olive plant is one of the herbal stimulants. Olive (*Olea europaea* L.) is an evergreen shrub or tree native to the Mediterranean region, belonging to the family Oleaceae [24]. Due to the health and nutritional benefits of the fruit, leaves, and oil, olives, have been utilized in traditional medicine since ancient times [25,26,27]. In traditional medicine, olive has been used for its antihypertensive, antiatherosclerotic, laxative, antipyretic, antiheadache, and energizing properties [28,29]. Numerous investors are drawn to the olive oil extraction industry by its multiple uses. However, the extraction process of olive oil generates numerous waste products, including olive leaves (OL), crude olive cakes (COC), and olive mill wastewater (OMW). Utilizing agro-industrial biomass offers excellent alternatives for developing fresh approaches to current and future aquaculture challenges as well as new phytogens capable of enhancing fish health and welfare. Additionally, polyphenols, flavonoids, iridoids, and carbohydrates that make up the majority of the olive leaf extract (OLE) [26] can function as antioxidants [30] and antibacterial [31] and antiviral agents [32]. The OLE has also been shown to have positive effects on fish disease resistance, expression of cytokine gene, and humoral immune responses [26,33].

The common carp (*Cyprinus carpio*) is the key cultured freshwater fish species contributing to the world economy [34], with an estimated 4.181 million tons yield worldwide in 2021 [35]. In addition to its low cost and superb nutritional content, carp has substantial marketability and economic [36,37]. The intense rearing practices used on the common carp in recent years may trigger stress and impair their immune systems, making them more vulnerable to diseases [38]. One of the most investigated bacterial infections linked to the common carp’s cytokine responses is *Aeromonas hydrophila* [39]. *A. hydrophila* infection is commonly followed by mass mortalities in aquaculture, which leads to significant economic losses [40]. Zoonotic illnesses are brought on by the food-borne pathogen *Aeromonas hydrophila* [41]. Localized pathological lesions may develop in the host tissue as a result. The host’s response could be manifested as tissue growth, ageing, and inflammation [42]. In addition to fish, amphibians, and reptiles, *A. hydrophila* is also a pathogenic agent for mammals, including humans, especially those in contact with sick fish [42,43]. The kidney, liver, gills, stomach, and spleen are among the organs in which degenerative histological alterations are frequently seen. Similar clinical symptoms and histological manifestations have been noted in recent studies of other *Aeromonas* species, including *A. jandaei* and *A. veronii* [44]. To combat this infection, phytochemicals might be an excellent alternative. To determine the optimum way to utilize the OLE as an immunostimulant for the *C. carpio,* more information is required regarding the effects of the OLE on the growth performance and the immunological response of the *C. carpio*. Therefore, the purpose of this study was to highlight the effects of the OLE on the growth performance and immunohematological, antioxidant, and histopathological changes in the *C. carpio* both under normal conditions and after the infection with *A. hydrophila*, as well as the mRNA expression of some growth-, antioxidant-, and immune-related genes such as *igfbp*, *sod*, *il1β*, *tnfα*, *caspase-3*, *nrf2*, *keap1*, and *pkc*.

## 2. Materials and Methods

Solvents used were of analytical or HPLC grade and were purchased from Merck (Darmstadt, Germany). The ^1^H and ^13^C NMR analyses were run on a Bruker AMX-500 NMR spectrometer (Karlsruh, Germany), operating at 400 MHz for ^1^H and 125 MHz for ^13^C NMR, respectively. DMSO_-_*d*_6_ as solvent and tetramethyl silane (TMS) as internal standard were used. HR-ESIMS analysis was carried out with an LCT Premier XE Micromass Waters spectrometer in the positive and negative ionization modes (Waters Corporation). Detection of compounds was carried out on precoated silica gel 60 F_254_ plates (0.25 mm layer, E. Merck, Darmstadt, Germany), using chloroform–methanol–water (8:2:0.2) as a mobile phase, and visualized with 5% H_2_SO_4_ spray reagent after heating at 110 °C.

### 2.1. Ethical Approval

Following the normal operating procedures approved by the Institutional Animal Care and Animal Ethics Committee, Faculty Aquatic and Fisheries Sciences, Kafrelsheikh University, Egypt (KFS-2020/8), the current experiment was conducted on common carp (*Cyprinus carpio*).

### 2.2. Plant Materials and Extraction

The leaves of *Olea europaea* L. were collected from the plant cultivated at the experimental station of the agriculture research center in October 2020. The plant was identified by Dr. Amr Salah, associate professor at Olive Research Department, Agriculture Research Center. A sample (OE 10.10.2020) was kept at the Herbarium of the Department of Pharmacognosy, Faculty of Pharmacy, Cairo University. The plant materials were air-dried and subjected to grinding; then, kept in dark airtight closed containers until the extraction step. Powdered leaves of *O. europaea* (500 g) were extracted with 60% ethanol (3 × 1500 mL), using Ultra-Turrax T25 homogenizer. The solvents were distilled under reduced pressure to give 78 g of yellowish-brown extract then lyophilized and kept at 4 °C for further biological tests.

### 2.3. Phytochemical Characterization and HPLC Analysis of Olive Leaf Extract

#### 2.3.1. HPLC Analysis of OLE

The OLE (546 mg) was diluted in 100 mL of a 50% methanol/water solution and sonicated for 20 min. The HPLC technique created by Savournin et al. [45] was adopted after a few minor changes. An Agilent HP1200 system with a G1322A quaternary pump, degasser, and photodiode array detector was used to conduct the HPLC analysis (PDA). The flow rate was maintained at 1.0 mL min^−1^ and UV monitored at 320, 280, and 254 nm. Chromatographic separations were performed on an Agilent ZORBAX Eclipse XDB-C18 column (250 × 4.6 mm i.d., 5 μm), including the C-18 guard column. Prior to HPLC analysis, all samples were filtered using Millex-HV filters (Millipore, Bedford, MA, USA) with 0.45 mm pore size. Injection volume of 20 μL was selected for the standardization process as an external standard. A standard calibration curve was established in the linear range of the detector from 100 to 600 μg mL^−1^. Under the previous condition, the oleuropein standard was eluted at 12.65 min, monitored, and analyzed at wavelength 280 nm.

#### 2.3.2. UPLC-ESI-MS/MS Analysis for Metabolite Analysis

The sample was analyzed by XEVO TQD triple quadrupole mass spectrometer (Waters Corporation, Milford, MA 01757 USA). The column used is ACQUITY UPLC–BEH C18 1.7 µm–2.1 × 50 mm column. HPLC grade solvents were used, and the system consisted of 0.1% formic acid in (A) water and (B) methanol. The gradient flow rate of 0.2 mL min^−1^ was 0 min, 90% A; 5 min, 70% A; 15 min, 30% A; 22 min, 10% A; 25 min, 10% A; 26 min, 100% B; 29 min, 100% B; 32 min, 90% A; followed by holding the initial conditions for 3 min for re-equilibration. The sample was dissolved in HPLC methanol, degassed, filtered, and then injected (10 μL) to LC-ESI-MS. Maslynx 4.1 software was used for data processing. Compounds were tentatively identified based on the comparison of the data with free databases and the published literature.

#### 2.3.3. Isolation and Purification of Oleuroside-10-Carboxylic Acid (I) and Demethyl Oleuroside-10-Carboxylic Acid (II)

The extract residue (900 mg) was resuspended in 50% methanol/water (100 mL) and extracted with an equal volume of ethyl acetate. The ethyl acetate fraction was evaporated under vacuum to yield a brown residue (290 mg). The obtained residue was chromatographed onto Sephadex LH20 (Sigma Aldrich Co., St Louis, MO, USA) column, and fractions of 1 mL were collected. Fractions of similar TLC (Thin Layer Chromatography) spots were pooled. Fractions 33–37 gave a yellow powder (8 mg) of I, while fractions 42–43 produced a yellow powder (5 mg) of II.

### 2.4. Fish, Diet, and Experimental Design

Fish were brought to the laboratory of the Fish Processing and Biotechnology Department, Faculty of Aquatic and Fisheries Sciences, Kafrelsheikh University from a private farm in Kafrelsheikh, Egypt. The fish were acclimated to the environment and fed a commercial meal ad libitum for 14 days (Figure 1). Then, fish with an average body weight of 9.24 ± 0.15 g, *n* = 180, were randomly stocked at a density of 15 fish per aquarium into 12 well-prepared glass aquariums (40 × 60 × 70 cm) (four treatments with three replications and 15 fish per each aquarium). The control and three test diets were prepared as shown in Table 1. The control group’s diet contained 0% of the olive leaf extract (0 OLE), and the three test diets contained the olive leaf extract (OLE) added at different levels—0.1, 0.2, and 0.3%. The fish were fed 2% of their body weight, twice daily, at 8:00 and 16:00, for 60 days. During the trial period, the water quality parameters were as follows: dissolved oxygen (DO) 6.5 0.5 mg L^−1^, pH 7.1 ± 0.8, electrical conductivity (EC) 219 ± 2 mho cm^−1^, temperature 23 ± 2 °C with day-and-night photoperiod 12:12 h. To maintain the water quality from declining during the trial, experimental fish were routinely monitored, and dead fish were removed. Fish in each tank were weighted individually at the beginning for recording IW (g) then biweekly and at the end of the experiment, using the tricaine methanesulfonate (MS-222) (100 mg L^−1^) to let them calm down. Growth parameters and feed utilization including final body weight (FBW), weight gain percentage (WG%), specific growth rate (SGR), and feed conversion ratio (FCR) were calculated for each treatment as follows:Weight gain (WG%) = [(FW − IW)/IW] × 100
Specific growth rate (SGR) = [(Ln FW − Ln IW)/t] × 100
Feed conversion rate (FCR) = feed intake (g)/weight gain (g)
Survival (%) = (final number of remaining fish/initial number of fish) × 100
IW = initial body weight (g), FW = final body weight (g), and t = the experiment duration in days

### 2.5. Challenge with A. hydrophila

After the 60-day feeding trial, fish from diet groups (control, OLE 0.1%, 0.2%, and 0.3%) were inoculated with PCR-identified pathogenic *Aeromonas hydrophila*, obtained from the Microbiology Research Center, Cairo University [46,47]. Initially, a pure culture of *A. hydrophila* was grown on trypticase soy agar (TSA) at 28 °C for 24 h, then a colony of *A. hydrophila* was carefully chosen and incubated on 10 mL of tryptic soy broth for 18 h in a shaking incubator. A tenfold serial dilution was prepared after the culture reached an OD_600_ of 0.6, then 1 mL of each dilution was evenly spread on TSA plates. Three replicates of each dilution were plated on TSA and incubated overnight at 28 °C. Importantly, CFU mL^−1^ was calculated using the dilution that resulted in 30–300 colonies. 

A preliminary estimation of LD_50–96h_ was performed using bacterial concentrations of 10^6^, 5 × 10^6^, 10^7^, 5 × 10^7^, 10^8^, 5 × 10^8^, and 10^9^ CFU mL^−1^ intraperitoneally injected to fish in a preliminary trial. Regression-based analysis of the mortality curves after challenge, in the preliminary trial, indicated that the estimated LD_50–96h_ was 1.62 × 10^8^ CFU mL^−1^. For the challenge trial, we selected a sublethal dose of 1/10 LD_50–96h_ (1.6 × 10^7^ CFU mL^−1^) [48]. Each fish in the challenged groups (20 fish/group) was intraperitoneally injected with 0.1 mL of the bacterial suspension. Simultaneously, fish from the same diet groups (20 fish/group) were inoculated with the same volume of sterile phosphate buffer saline to neutralize the effect of injection process during the two weeks of mortality monitoring. Infection was confirmed by re-isolation of inoculated bacteria from challenged fish by the end of the challenge, and pathogenicity was confirmed based on PCR amplification of the *A. hydrophila* aerolysin-A toxin gene (For. TTGACCTCGGCCTTGAACTC, Rev. GTGAAACCGAACTGGCCATC, NCBI accession number: KX138395.1). The challenge trial lasted for 15 days with daily monitoring and mortality reporting.

### 2.6. Collection of Blood Samples and Tissue Specimens

Five fish per replicate (15 fish per treatment) were randomly chosen after the feeding trial (8 weeks) and five days after the bacterial challenge. They were initially given a mild anesthetic, using (MS-222) 100 mg L^−1^ (tricaine methane sulfonate, Sigma, WA, USA). Each fish had two blood samples taken from the caudal vein. The first portion was put into tubes that had been heparinized to act as an anticoagulant for hematological testing and phagocytic activity, while the remaining blood was stored in plain tubes and centrifuged for 15 min at 3000× *g*, 4 °C, to obtain serum. The serum was divided into aliquots and kept at −80 °C for additional investigation. After blood was drawn from the fish, tissue samples of the intestine, hepatopancreas, spleen, and kidney were taken. Hepatopancreas samples were used for the antioxidant and qRT-PCR assays and were held at −80 °C for total RNA separation. They were kept in a sterile microcentrifuge tube. The other tissue samples were collected in 10% neutral buffered formalin as a fixative solution and used for histopathological analysis. Intestine samples (anterior part) were used for the assessment of the digestive enzymes. 

### 2.7. Assessment of the Immunohematological Parameters

Red blood cells (RBCs) and white blood cells (WBCs) were counted using a Neubauer hemocytometer with Natt-Herring fluid as stated by Natt and Herrick [49]. The differential leukocytic count was also evaluated. Hematocrit value was calculated using the microhematocrit technique, while the hemoglobin was measured using the conventional cyanmethemoglobin technique according to Stoskoph [50]. The results were used to calculate the RBC indices, including mean corpuscular volume (MCV), mean corpuscular hemoglobin (MCH), and mean corpuscular hemoglobin concentration (MCHC). *Candida albicans* was used to perform phagocytic activity (PA) of polymorphonuclear cells, and these procedures were performed in accordance with the instructions provided by [51]. 

### 2.8. Assessment of the Serum Biochemical and Immunological Parameters

Total proteins (TP) and albumin (Alb) were examined in the serum samples, according to Burtis and Ashwood [52] and Dumas and Biggs [53], respectively. Alanine amino transferase (Alt), aspartate amino transferase (Ast) serum enzyme activities were measured in accordance with Reitman and Frankel [54]. Total cholesterol (TC) and triglycerides (TG) were assessed according to Richmond [55], while urea and creatinine were determined according to Henry et al. [56] and Szasz et al. [57], respectively. All these parameters were assessed. The diagnostic reagent kits for spectrophotometer (MyBioSource Inc., San Diego, CA, USA) were used in according to the manufacturer’s instructions. The amount of globulins (Glob) was determined using the procedure of Kaneko [58]. The method described by Demers and Bayne [59] to measure the lysozyme activity of sera, using an ELISA microplate reader. The method, Sigma-Aldrich, 500 mg L^−1^ of *Micrococcus lysodeikticus*, was designed to assess the diameter of clear lysed zones formed by various serum samples on 1% agarose gel and then contrasted with those produced by the reference solution (20 mg mL^−1^ hen egg-white lysozyme). In order to do this, serum samples (25 L) were inoculated in a 50 mM phosphate buffer (1%) with agarose gel (pH 6.3). The plates were then securely fastened and maintained at 37 °C for 18 h. The formula Y = A + Blog X, where X is the serum lysozyme activity expressed in g/mL, and Y is the width of the lysed zone, was used to assess the serum lysozyme activity. Utilizing an ELISA kit specific to fish IgM, the serum immunoglobulin (IgM) concentration was assessed (Cusabio and Cusab, Houston, TX, USA). Using the nitroblue tetrazolium (NBT) assay, blood-wide respiratory burst activity was measured according to Secombes [60]. The NBT drop at 630 nm was measured using a microplate reader (Optica, Mikura Ltd., Pocklington, York, UK).

### 2.9. Estimation of the Intestinal Digestive Enzyme

Intestinal samples (*n* = 15/group) were homogenized by an electric homogenizer on ice before analysis. For homogenization, the manufacturer’s instructions for the commercial kit were followed using the specific buffer for each enzyme and centrifuging the mixture for 30 min at 3000× *g*, 4 °C. Before analysis, the supernatants were kept at −80 °C. Measurements of protease: protease, amylase, and lipase activities were carried out as described in [61,62,63]. Alkaline phosphatase was measured as described and modified by Bassey et al. [64] and Wright et al. [65], respectively.

### 2.10. Estimation of the Hepatic Antioxidants and Lipoperoxidation Biomarkers

Dissected liver tissues were washed in a solution containing ice, 50 mM sodium phosphate buffered saline (100 mM 3 Na_2_HPO_4_/NaH_2_PO_4_, pH 7.4), and 0.1 mM EDTA to eliminate any RBCs and clots. The tissues were then centrifuged at 5000 rpm for 30 min after being homogenized in 5–10 mL of cold buffer per g of tissue. The resultant supernatant was put into an Eppendorf tube and divided into aliquots to be stored at −80 °C. Using thiobarbituric acid, the quantity of hepatic malondialdehyde (MDA) was measured [66], superoxide dismutase (Sod) [67], catalase (Cat) [68], and glutathione peroxidase (Gpx) [69] were assessed spectrophotometrically according to the manufacturer’s instructions, using diagnostic reagent kits (MyBioSource Inc., San Diego, CA, USA).

### 2.11. Histopathological Examination

Five fish were chosen at random from each fish group after eight weeks of feeding trials and five days of being exposed to *A. hydrophila* for a histopathological analysis. The abdomen was dissected after being deeply antisepticized with 70% ethyl alcohol, and tissue samples from the hepatopancreas, spleen, kidney, and middle part of the small intestine were preserved for at least 24 h in 10% neutral buffered formalin. The tissue samples were cleaned in xylene, embedded in paraffin wax, and dehydrated using increasing concentrations of ethanol (70–100%). Leica RM 2125 microtome (Leica DM 5000, Leica Biosystems, Richmond, IL, USA) was used to slice tissues into 5 mm thick slices, which were subsequently stained with hematoxylin and eosin (H&E) and viewed under a light microscope (Leica DM 5000).

### 2.12. Total RNA Extraction, cDNA Synthesis and Real-Time Quantitative PCR Assay

This work was performed in the Biotechnology Lab, Faculty of Aquatic and Fisheries Sciences, Kafrelsheikh University. To evaluate the hepatic growth, immunity, and antioxidant genes expression levels, 5 fish/replicates were collected from all tested groups in 2 mL sterile Eppendorf tubes and immediately shocked in liquid nitrogen and kept at −80 °C until the time of further use for RNA extraction. Total RNA was extracted from 50 mg hepatic tissues, using Trizol (iNtRON Biotechnology) and following the manufacturer’s instructions. The quality and quantity of the extracted RNA were evaluated using ethidium-bromide-stained 2% agarose gel electrophoresis. RNA concentration was assessed by NanoDrop^®^ BioDrop Spectrophotometer (ThermoFisher Scientific, Carlsbad, CA, USA). Two µg of extracted RNA sample was reverse-transcribed using Maxime RT PreMix (Oligo dT primer) (iNtRON Biotechnology, Seongnam-Si, Republic of Korea) and following the manufacturer’s manual. Gene expression analysis was performed in the Mic Real-time PCR system (Bio-molecular systems, Upper Coomera, QLD, Australia), using the SensiFast SYBR No-Rox kit (Bioline) with the common carp’s gene-specific primers [70,71,72,73,74] (Table 2). Real-time PCR amplifications, using SensiFast SYBR Lo-Rox kit (Bioline, UK), were carried out in a total volume of 20 μL reaction mixtures that contained 2 μL of cDNA, 0.5 μL of each primer, and 10 μL of SeniFast^TM^ SYBR Lo-Rox master mix (Bioline, London, UK). The amplification cycling conditions were as follows: initial denaturation at 95 °C for 10 min, followed by 40 cycles at 95 °C for 15 s, 60 °C for 1 min and 72 °C for 30 s. Melting curve analyses were executed to confirm the specific amplification of each target gene. The relative expression folds of each gene were calculated according to 2^−ΔΔCT^ method and corrected amplification efficiencies [75,76], using β-actin as an internal control for standardizing the expression of other genes. 

### 2.13. Statistical Analysis

Data are expressed as mean ± standard error of the mean. Prior to analysis, Shapiro–Wilks and Levene’s tests were used to check the data distribution normality and homogeneity (*p* < 0.05), respectively. Growth performances and the relative gene expression data were analyzed using one-way ANOVA, whereas pre- and post-challenge hematological data, serum biochemistry, serum immune parameters, intestinal enzyme activities, and hepatic antioxidants activities were subjected to two-way ANOVA analysis, each followed by Tukey’s multiple comparison as a post hock test (*p* < 0.05) to evaluate the differential response of experimental groups to OLE dosages in terms of mean differences. Kaplan–Meier regression curves with log-rank test were conducted to analyze the cumulative survival data during the two-week challenge trial (*p* < 0.05). To determine the best-performing OLE dosage, growth and immune performances data were interpolated using polynomial regression models [76,77]. Adjusted-R2, AICs, and sum of squares criteria of polynomial models were compared to choose the best fitting model for each interpolated parameter. Quadratic polynomial model (second-order) was conducted to interpolate the best-performing FCR dosages, while the cubic polynomial (third-order) regression model was conducted to interpolate the optimum-performing OLE dosages for WBC count, phagocytic activity, Sod activity, and Lysosome activity [76,77]. The statistical analyses were conducted using GraphPad Prism (version 9.1, GraphPad Software, San Diego, CA, USA).

## 3. Results

### 3.1. Phytochemical Characterization of the OLE 

The HPLC analysis revealed the presence of oleuropein as the main phenolic compound of the *O. europaea* leaf extract at a concentration of 20.4%. The UPLC-ESI-MS/MS analysis revealed 46 peaks (Appendix A), which were tentatively identified in both positive and negative ion modes (Appendix A) and isolated compounds I and II (Appendix A). The results were consistent with the data in the literature [78,79,80,81,82,83,84,85]. The Secoiridoids detected were oleuropein and its hydroxy, dimethyl, demethyl, dihydro and dehydro derivatives, ligustroside, oleuroside, neonuzhenide, verbasoside, verbascoside, isoverbascoside, jaspolyoside, oleoside, and secologanic acid. In addition, loganic acid and mono- and diglycosides of elenolic acid were also identified. The flavonoids detected were gallocatechin, glycosides of quercetin, luteolin and apigenin. The identified phenolics were protocatechuic acid, caffeic acid, hydroxy benzoic acid, *p*-coumaric acid, tyrosol, and hydroxytyrosol. Other compounds found were the lignan acetoxypinoresinol, triterpenoid oleanolic acid, the anthocyanin cyanidin-3-*O*-rutinoside, and citric and gluconic acids.

#### 3.1.1. Identification of Secoiridoids in OLE

Oleuropein, the main secoiridoid in the OLE, exhibited an ion peak at *m*/*z* 539 for [M − H]^−^. The MS/MS fragment at *m*/*z* 377 is characteristic for oleuropein aglycon. Oleuroside had the same mass and MS/MS fragment ions as oleuropein; however, it eluted later at retention time (7.28 min). Neonuzhenide showed MS/MS fragments similar to oleuropein and oleuroside; however, it had an [M − H]^−^ ion at *m*/*z* 701. Hydroxy oleuropein showed an ion peak at *m*/*z* 555, which is 16 Da extra to that of oleuropein, and its MS/MS fragment at *m*/*z* 537 was corresponding to the loss of water. Demethyl oleuropein, with an ion peak at *m*/*z* 525, had 14 Da less compared to oleuropein due to the loss of a methyl group. Dihydro oleuropein was identified by its ion peak at *m*/*z* 543, which is 4 Da extra to oleuropein. Dimethyl oleuropein aglycon exhibited an [M − H]^−^ at *m*/*z* 405 with 28 Da extra to oleuropein aglycon (*m*/*z* 377) accounting for two methyl groups. Verbasoside showed [M − H]^−^ and [M + H]^+^ ion peaks at *m*/*z* 461 and 463, respectively. The MS/MS fragments at *m*/*z* 315, 297, and 135 indicated the subsequent loss of rhamnosyl, water, and hexosyl molecules, respectively. Verbascoside and isoverbascoside had an [M − H]^−^ ion peak at *m*/*z* 623. The fragmentation pattern is the same; however, they eluted at different retention times (6.10 and 6.35 min, respectively) as reported in the literature [86,87,88]. Oleoside methyl ester showed its characteristic [M − H]^−^ ion peak at *m*/*z* 403, which by the loss of [OCH_3_] produced an ion peak at *m*/*z* 371 for oleoside-H_2_O. Ligustroside and jaspolyoside exhibited [M − H]^−^ ion peaks at *m*/*z* 523 and 925, respectively. Their respective MS/MS fragmentation profiles were similar to those published. Secologanic acid was tentatively identified from its mass in the positive ion mode at *m*/*z* 425.

#### 3.1.2. Identification of Flavonoids in OLE

Luteolin diglucoside and luteolin hexoside were tentatively identified by their [M − H]^−^ ion peaks at *m*/*z* 609 and 447, respectively. Additionally, the loss of 162 Da for hexosyl parts resulted in the formation of the molecular ion at *m*/*z* 285, which is characteristic for luteolin.

Apigenin-*O*-rutinoside and apigenin-*O*-glucoside exhibited [M + H]^+^ and [M − H]^−^ ion peaks at *m*/*z* 579 and 431, respectively. Loss of 132 Da and 162 Da for rhamnosyl and glucosyl parts, respectively, resulted in an ion peak at *m*/*z* 271 [M + H]^+^ and at *m*/*z* 269 [M − H]^−^, which represented the aglycon apigenin.

Rutin and quercitrin showed molecular ion [M − H]^−^ at *m*/*z* 609 and 447, respectively. Both glycosides, by the loss of the sugar part, yielded an ion peak at *m*/*z* 301, which is characteristic for the aglycon quercetin.

Gallocatechin was identified by its [M − H]^−^ ion peak at *m*/*z* 305, and the MS/MS fragments were typical compared to the literature.

#### 3.1.3. Identification of Phenolics in OLE

Protocatechuic acid exhibited an [M − H]^−^ ion peak at *m*/*z* 153 and the characteristic MS/MS fragments at *m*/*z* 141 and 109. Caffeic acid showed its [M − H]^−^ ion peak at *m*/*z* 179, which yielded an MS/MS-fragment ion at *m*/*z* 135. Both hydroxybenzoic acid and tyrosol showed an [M − H]^−^ ion peak at *m*/*z* 137. However, both exhibited their characteristic relevant MS/MS pattern in which hydroxybenzoic acid gave an MS/MS fragment at *m*/*z* 109 while tyrosol was fragmented to the ion at *m*/*z* 107. Hydroxytyrosol was characterized by its ion at *m*/*z* 153, which is 16 Da extra to tyrosol. Hydroxytyrosol hexoside gave an ion peak at *m*/*z* 315, which by fragmentation lost 162 Da for hexosyl part and yielded a fragment ion at *m*/*z* 153 for tyrosol. The *p*-coumaric acid had an [M − H]^−^ ion peak at 163 and was fragmented to an ion at *m*/*z* 117.

#### 3.1.4. Identification of Unknown Compounds

Unknown compounds with mass adduct at *m*/*z* 569 in negative ion mode and 607 in positive ion mode were detected. The mass fragments at *m*/*z* 541, 539, and 377 indicated that they could be derivatives of oleuropein or oleuroside. These compounds were targeted for isolation and spectral identification. 

#### 3.1.5. Structure Elucidation of Compounds I and II

The ESIMS of I showed an adduct ion at *m*/*z* 607.1639, which, with the ^13^C NMR data, indicated a molecular formula of [C_26_H_32_O_15_ + Na]^+^ (calculated 607.1638). The ^1^H NMR chemical shifts of compound I (Appendix A) indicated three olefinic aromatic protons at δ_H_ 6.64 for H-4′, δ_H_ 6.92 for H-7, and δ_H_ 7.47 for H-8′. A doublet for an anomeric proton at δ_H_ 4.7 (*J* = 8 Hz) and the chemical shifts at δ_C_ 103.7, 77.6, 75.3, 73.6, 69.9, 60.9 (Appendix A) indicated a β-glucosyl moiety. The signal at δ_H_ 6.77 was assigned for H-5. Two doublets at δ_H_ 2.34 and 2.68, a multiplet at δ_H_ 1.25 for H-3, a multiplet at δ_H_ 0.9 for H-7, two multiplets at δ_H_ 3.72 and 3.5 for H1′, a multiplet at δ_H_ 2.6 for H-2, and two doublets at δ_H_ 7.43 and 6.80 for H-8 and H-9, respectively, proposed a secoiridoid structure of the oleuroside type [89] substituted at C-9. Additionally, a singlet at δ_H_ 13.00 indicated an acid and a singlet at δ_H_ 3.20 for a methyl ester with δ_C_ 56.6. The ^13^C NMR data indicated the presence of three carbonyls at δ_C_ 161.6, 161.5, and 163.8. The comparison of these data with the published literature [89] proposed that compound I is oleuroside-10-carboxylic acid. This is the first report for the isolation and NMR spectral identification of this compound. It was previously tentatively identified by LCMS only [81].

Compound II showed spectral data comparable to those of compound I (Appendix A). However, its mass adduct ion at *m*/*z* 569.1507 alongside the NMR data indicated a molecular formula of [C_25_H_30_O_15_ − H]^−^, implying a difference in a methyl group compared to compound I. The singlet at δ_H_ 3.2 for a methyl ester in compound I was found absent in compound II with the presence of two singlets at δ_H_ 13.0 and 13.1. These differences suggested a demethyl form of compound I. Compound II was identified as a demethyl oleuroside-10-carboxylic acid which, to our knowledge, was not isolated or identified from any source yet, which requires full spectral identification of the new compounds in future studies. NMR and HRESIMS spectra of the isolated compounds are included in the Appendix A. 

### 3.2. Growth Performance

The growth responses of the *C. carpio* fed the test diets containing graded doses of the OLE (0, 0.1, 0.2, and 0.3%) are summarized in Table 3. Fish growth traits (FBW, WG%, and SGR values) were the highest; furthermore, FCR was the best (*p* < 0.05) in the fish group fed the diet supplied with the 0.1% OLE compared to the control and the other groups. Fish growth markedly lowered (*p* < 0.05) as the dietary OLE dose increased to 0.2% and 0.3%. The best growth-promoting dose was 0.066, as revealed in the FCR interpolation from the standard curve (Figure 2).

### 3.3. Disease Resistance

The survival rate of the *C. carpio* fed the OLE-supplemented diets two weeks after the exposure to the *A. hydrophila* challenge is presented in Figure 2. The data revealed that the 0.1% OLE had a noticeably higher survival rate compared to the infected control group and the groups supplied with higher OLE concentrations (0.2 and 0.3%) (*p* < 0.05). The mortality of the infected unsupplied group and the 0.3% OLE fed group started following the second day after the exposure to *A. hydrophila*, while in the fish fed the 0.2% OLE, it started after the third day. Moreover, the mortality of the 0.1%-OLE-fed group started after the fourth day. 

### 3.4. Hematological Findings

The effects of various dietary OLE concentrations on the *C. carpio*’s hematological parameters both before and after the infection with *A. hydrophila* are outlined in Table 4.

Regarding the erythrogram parameters, RBCs, PCV, and Hb did not differ significantly between the dietary treatments (*p* > 0.05). Furthermore, only the Hb parameter showed a significant increase after the *A. hydrophila* challenge (*p* > 0.05).

By increasing the dose of the dietary OLE in a dose–response way in comparison to the control group, there was a notable rise in basophil counts (Table 4). Additionally, when compared to the control noninfected group, the *A. hydrophila* challenge considerably increased the WBC, heterophil, monocyte, and basophil counts while significantly reducing the lymphocyte counts. Moreover, the OLE supplementation was more effective at reducing the effects of the *A. hydrophila* challenge on leukogram parameters than the control challenged group (*p* < 0.05). Importantly, the best-performing OLE dose for the WBC-count, based on the interpolation from the standard curve, was 0.076% (Figure 3).

### 3.5. Serum Biochemical and Immunological Parameters

The effects of the dietary OLE and/or *A. hydrophila* infection on the serum biochemical measurements of the *C. carpio* are shown in Table 5.

The dietary inclusion of the OLE significantly enhanced the total protein and globulin concentrations compared to the control before and after the bacterial challenge with the best enhancement with the 0.1% OLE compared to the control group.

Moreover, serum enzyme activities ALT and AST were similar in the 0.1% OLE and control group before the bacterial challenge. However, after the challenge, the fish fed the 0.1% OLE exhibited the lowest values of these biomarkers of liver dysfunction compared to the other groups (*p* < 0.05), as shown in Table 5.

Regarding the serum cholesterol levels shown in Table 5, nonsignificant changes were observed in the fish group fed the 0.1% OLE compared to the control. However, the levels of triglycerides were significantly lower in the 0.1% OLE group compared to the other treatments, both pre- and post-challenge (*p* < 0.05). Regarding serum kidney injury biomarkers, BUN concentration significantly declined in the fish group fed 0.1% OLE, while it was significantly elevated in the other groups in a dose–response manner compared to the control group either before or after being bacterially challenged. Moreover, serum creatinine levels were significantly elevated by the increased dietary dose of the OLE compared to the control group either pre- or post-challenge.

Furthermore, feeding the dietary 0.1% OLE to the *C. carpio* remarkably enhanced the measured immunohematological parameters including phagocytic activity (PA%), lysozyme (LYZ), and burst activity (NBT) following the *A. hydrophila*-challenging conditions (Table 5). On the other side, feeding higher doses of the OLE (0.2 and 0.3%) either before or after the *A. hydrophila* challenge exhibited a significant reduction in the aforementioned immune parameters compared to those of the control group (Table 5). Importantly, the best-performing OLE dose for the WBC count, phagocytic activity%, and lysozome activity, based on the interpolation from the standard curve, was 0.76, 0.74, and 0.74 g kg^−1^ diet, respectively (Figure 3).

### 3.6. Hepatic Antioxidants and Intestinal Digestive Enzymes and Intestinal ALP

Changes in the hepatic antioxidant (Sod, Cat, and Gpx) and oxidative stress biomarker (MDA) as well as the intestinal digestive enzymes and intestinal ALP in the *C. carpio* fed the OLE either before or after the *A. hydrophila* challenge are described in Table 5. The activities of the hepatic antioxidant enzymes, Sod and Gpx, were remarkably increased in the group supplied with the 0.1% OLE, while they were significantly inhibited by the increased dietary dose of the OLE in a dose–response manner either pre- or post-infection compared to the control group. On the contrary, the serum MDA levels showed the opposite trend to the antioxidant enzyme activities in the group supplied with the 0.1% OLE, which were significantly enhanced by the increased dietary dose of the OLE in a dose–response manner either pre- or post-infection compared to the control group (Table 5).

Interestingly, the activity of intestinal enzymes, including α-amylase, protease, and lipase, as well as intestinal Alp of the *C. carpio* fed different dietary OLE concentrations, is shown in Table 5. A notably higher enzymatic activity of protease and Alp was observed in fish fed the 0.1% OLE compared to the control, both before and after the bacterial challenge (*p* < 0.05). Regarding the activity of amylase and lipase, the fish fed the 0.1% OLE exhibited the highest activity before the challenge (*p* < 0.05) for amylase and after the challenge for lipase (*p* < 0.05). However, the *A. hydrophila* infection significantly inhibited their activities compared to the control uninfected group. Moreover, their activities also remarkably declined by the increased dietary dose of the OLE in a dose–response manner compared to the control group either pre- or post-challenge (Table 5).

### 3.7. Histopathological Observations

Histopathological changes in the middle part of the intestine, hepatopancreas, spleen, and kidney for different groups are shown in Figure 4, Figure 5, Figure 6 and Figure 7. The middle part of the intestine of the normal control healthy *C. carpio* fed the basal diet without the OLE showed intact intestinal villi lined by simple columnar epithelium with goblet cells and lamina propria of loose connective tissue (Figure 4A). Moreover, dietary inclusion of the 0.1% OLE showed increased length and branched intestinal villi (Figure 4B). Dietary inclusion of the 0.2% OLE revealed degeneration and sloughing of the apical part of villi (Figure 4C). Similarly, the dietary inclusion of the 0.3% OLE showed edema and degenerative changes in the lamina propria with degeneration and sloughing of the apical part of villi and lymphocytic aggregations (Figure 4D). Moreover, the group infected with *A. hydrophila* revealed the degeneration and sloughing of the lining epithelium of the intestinal villi with edema in the lamina propria in addition to marked lymphocytic aggregations besides the presence of hemorrhage (Figure 4E). 

The hepatopancreas of the normal control unsupplied group showed normal architecture of liver with intact central vein and polyhedral-shaped hepatocytes separated by blood sinusoids with normal hepatic architecture (Figure 5A). The fish fed the 0.1% OLE showed mild vacuolar changes in hepatocytes and pancreatic acini (Figure 5B), while the fish fed the 0.2% OLE showed moderate vacuolar changes in hepatocytes and congestion in both hepatic sinusoids and pancreatic blood vessels (Figure 5C). The fish fed the 0.3% OLE showed severe vacuolar degeneration in hepatocytes and congestion in hepatic sinusoids (Figure 5D). Moreover, the group infected with *A. hydrophila* revealed nuclear pyknosis, vacuolar degeneration, hemosiderosis, and congestion of the hepatic and pancreatic blood vessels (Figure 5E). The severity of lesion increased with increasing the OLE concentration in the fish ration (Figure 5G,H).

The spleen in the control group showed mixed white and red pulp, ellipsoids, melanomacrophage centers, and pancreatic acini (Figure 6A). The group fed the 0.1% OLE revealed normal splenic architecture: interconnecting cords of red pulp, ellipsoid arterioles, and melanomacrophage centers surrounded by white pulp of lymphoid cells in addition to exocrine pancreatic acini (Figure 6B). The increased OLE concentrations showed congestion of splenic blood vessels, degenerative changes, and lymphocyte depletion besides interstitial edema in a dose–response manner (Figure 6C,D). Moreover, the group infected with *A. hydrophila* showed vacuolar degeneration of the lymphoid elements surrounding the ellipsoid arterioles (Figure 6E). Histopathological changes in the spleen of the OLE-fed groups and infected with *A. hydrophila* showed edema in splenic parenchyma, degeneration of lymphoid elements, and mild increase in melanomacrophage centers in a dose-dependent increase in the lesion (Figure 6F–H). 

The kidney of the control and 0.1% OLE-fed groups showed intact renal glomeruli and tubules (Figure 7A,B) surrounded by mildly congested blood vessels and interstitial tissue containing normal hematopoietic tissue (Figure 7B). Fish fed the 0.2% and 0.3% OLE showed degeneration and sloughing of tubular epithelium, congestion of blood vessels, and hemosiderosis in a dose–response manner (Figure 7C,D). 

Furthermore, the group challenged with *A. hydrophila* revealed degeneration in the renal glomeruli, degeneration and separation of the tubular epithelium with the presence of hyaline cast, edema, and degeneration of the interstitial tissue in addition to hemosiderosis (Figure 7E). Histopathological changes in the kidney of the OLE-fed groups infected with *A. hydrophila* showed a wide capsular space of glomeruli, vacuolar degeneration of renal tubules, and edema and degeneration of hematopoietic tissue besides the presence of melanomacrophage centers in a dose-dependent increase in the lesion (Figure 7F–H).

### 3.8. mRNA Expression Profile

The effects of the dietary OLE treatments, together with experimental infection by *A. hydrophila,* on the transcriptional levels of liver growth-related (*igf-bp*), antioxidant-related (*sod*, *nrf2*, *pkc*, and *keap1*), and immune-related genes (*tnfα* and *il1β*) and apoptotic gene (*caspase-3*) of the *C. carpio* five days after the bacterial challenge are portrayed in Figure 8. Generally, there was a significant difference between the experimental groups for all the studied genes. 

Concerning the transcriptional levels of liver growth-related *igf1bp*, following the *A. hydrophila* infection, the dietary 0.1% OLE supplementation induced a marked decrease in *igf1bp*, while its expression levels increased with increasing the OLE incorporation dose in a dose–response manner compared to the control infected group (Figure 8A).

Regarding the transcriptional levels of hepatic antioxidant-related genes including *sod*, *nrf2*, *pkc*, and *kaep*1, the OLE feeding exhibited a higher antioxidant response via enhancing *sod*, *nrf2*, *pkc,* and *kaep1* transcription levels compared to the control unsupplied infected group. Interestingly, the maximum stimulation was noticed in the fish group at 0.1% of the OLE compared to the other groups (Figure 8C,F–H).

Concerning the relative mRNA expression levels of pro-inflammatory cytokines such as *il1β* and *tnfα*, as well as apoptotic marker *caspase-3*, the OLE supplementation at a dose of 0.1% downregulated their expression levels compared to higher doses (0.2 and 0.3%). The lowest expression level was exhibited in the fish group fed the 0.1% OLE post-infection compared to higher doses of the OLE as illustrated in Figure 8B,C,E. 

## 4. Discussion

In the green economy era, using plant extracts in aquaculture appears to be a sustainable and socially acceptable tactic [90]. Many other additives, such as probiotics, yeast, antioxidants, algae, and plant extracts, are frequently added to the diets of farmed fish to improve nutrient consumption, growth performance, appetite stimulation, and survival [10,11,13,91,92,93]. The OLE was effective in the current study in modulating *A. hydrophila* infection because it is a rich source of flavonoids and phenolics in addition to dimethyl oleuroside-10-carboxylic acid, which helps to increase the antioxidative response and stimulate the immune system [90,93,94].

In this study, we focused on evaluating the impact of the OLE on key points targeting growth performance, immunity, antioxidant capacity, mRNA expression of some antioxidant, inflammatory and apoptotic genes under normal conditions or after the *A. hydrophila* infection in the *Cyprinus carpio*. We found that 8 weeks of the 0.1% OLE dietary inclusion had a positive effect on the overall carp growth performance via modulating the FBW, BWG%, and FCR through enhancing intestinal digestive and ALP enzymes either before or after the *A. hydrophilas* infection compared to the results of the control and higher OLE doses (0.2% and 0.3%). The result was consistent with that of Baba et al. [26] and Zemheri-Navruz et al. [33,90], who confirmed improved digestibility of protein, lipid, and carbohydrate in fish fed 0.1% OLE by stimulating the intestinal digestive enzyme activities. Kaleeswaran et al. [95] stated the enhanced expression of growth-related genes (*gh* and *igf-1*) in the brain and liver of the common carp, besides improving the appetite and food consumption [96] in addition to the OLE’s capacity to promote the development of fish gut microbiota such as *Lactobacillus acidophilus* by excreting probiotic effectiveness [97]. On the other hand, we also noticed a weight reduction in the *C. carpio* given higher OLE dietary doses in a dose-dependent response manner. The OLE was found to contain phenolic compounds, mainly oleuropein and its derivatives. Shen et al. [98] studied the effect of the OLE on obesity and showed that this extract can modulate the expression of genes involved in adipogenesis and thermogenesis. Little scattered data in the literature were found regarding the anorexic effect of the phenolics of the OLE. The anorexic effect of the olive oil was attributed mainly to oleic acid which is converted to oleoylethanolamide. Oleoylethanolamide acts as a hormone and decreases the appetite [99]. Sato et al. [100] investigated the effect of oleanolic acid as an agonist for TGR5 receptor that stimulates the glucagon-like-peptide-1 (GLP-1) secretion. GLP-1 is an enteroendocrine hormone GLP-1 with important actions including reduction of food intake. In our study, oleanolic acid was detected in the OLE, which can exert the same effect on GLP-1. Thus, oleuropein and all its derivatives can be considered potential anorexic compounds [94,101]. Moreover, oleuroside and the new compounds—oleuroside-10-carboxylic and its dimethyl form, verbasoside, isoverbascoside, and verbascoside—have a structural similarity with oleuropein; therefore, they are anticipated to exert the same effect. However, more investigation is required.

Luteolin, apigenin, and flavone aglycones showed the activity of increasing GLP-1 secretion as reported in the literature [102]. Amongst the identified compounds in our study are glycosides of luteolin and apigenin, which by metabolism would generate the corresponding aglycones resulting in the same effect on GLP-1.

The *p*-coumaric acid, a phenolic compound detected in our study, showed the activity to ameliorate obesity through increasing the secretion of GLP-1 [103]. Cyanidin, an anthocyanin, was proved to increase the GLP-1 secretion [104]. Cyanidin-3-O-rutinoside, detected in our study, through metabolism losses of the sugar part and cyanidin would generate and exert a positive effect on the GLP-1 production.

The observed growth-retardation effect of higher doses of the OLE in our study could be explained by the consortium of the different constituents as discussed above. Here, we detected that the fish group fed the 0.1% OLE and challenged with *A. hydrophila* exhibited the highest intestinal ALP activities either before or after the bacterial challenge in comparison to the control group, indicating improved fish disease resistance and survival rate with a lower level of *igfbp* mRNA expression compared to the other higher doses, which may be explained by the increased nutrient utilization at this level, thus enhancing growth and excreting antibacterial effects, in agreement with Sudjana et al. [105], Gullón et al. [106], and Centrone et al. [107], who reported antibacterial efficacy of the OLE against *Staphylococcus aureus* and *Escherichia coli*. The increased ALP activities in the fish group treated with the 0.1% OLE in comparison to the control group strengthened the immune system’s response prior to or following the bacterial challenge and reduced the release of pro-inflammatory cytokines (*tnfα* and *il1β*) after the infection (Figure 8B,C). The results were supported by Yang et al. [108] and Wang et al. [109], who declared immunological roles of intestinal ALP in the dephosphorylation and detoxification of pro-inflammatory components, the inhibition of the synthesis of inflammatory proteins, and the enhancement of mucosal tolerance to Gram-negative bacteria in zebrafish [110]. Interestingly, the OLE’s phenolic compounds can eliminate pathogenic bacteria from the digestive organs [111]. Yilmaz et al. [93] observed that a combination of probiotics (*Bacillus subtilis*) and phenolic acid and trans-cinnamic acid increased the intestinal amylase activity and decreased the numbers of coliform and *Enterobacteriaceae* in the intestine of the rainbow trout. Therefore, impact the normal physiological condition, particularly the immunological indicators [112,113]. In the current study, we found that there was a direct decrease in all immunological indices (total proteins, globulins, phagocytic activity, lysozyme, and NTB and IgM concentrations) as well as a considerable elevation of WBCs, heterophils, and monocytes following the *A. hydrophila* infection. Additionally, the OLE dietary therapy was able to control this scenario by boosting intestinal mucosal immunity [114] on gilthead seabream (*Sparus aurata*). High antioxidant levels that reduce oxidative stress, which weakens the innate immune response in fish and increases susceptibility to disease, may be one of the reasons of these immune-enhancing effects of the OLE [115]. Gholamhosseini et al. [113] utilized the OLE against the white-spot-virus syndrome in shrimp (*Litopenaeus vannamei*) [116,117]. Similarly, Branciari et al. [118] emphasized the possible impact of the addition to poultry diet of dehydrated olive cake and olive polyphenol extract on the release of *Campylobacter* spp. Along the same lines, Zemheri- Navruz et al. [33] stated that the best OLE-supplying dose was 0.1% which boosted the immune parameters and increased the survival rate against *E. tarda* in the common carp fingerlings.

In addition to the aforementioned outcomes, the OLE therapy had no discernible impact on the parameters of the erythrogram as compared to the control group. In any case, RBCs, Hb, and PCV were marginally increased after exposure to *A. hydrophila*, but the effects were not statistically significant when compared to the control group. The increase in the RBC count may be due to the tissue hypoxia as a result of erythropenia caused by *A. hydrophila* infection, which was affected by the degree of hemolytic anemia resulting from potent *A. hydrophila* cytolysin which was capable of lysing RBCs [119,120]. Because of this, the tissues are not receiving enough oxygen, which causes cell release from the hemopoietic organs and an increase in the erythropoietin hormone synthesis [121], as a result of the increased supply of RBCs from the spleen too.

In this study, we found that WBC counts were significantly increased in all the three groups especially the group fed the 1% OLE. Zemheri-Navruz et al. [33] supported our findings that the optimal OLE-supplied dose to raise immunological parameters and increase the common carp fingerlings’ survival rate against *E. tarda* was 0.1% OLE. The leukocytic count acts as an indicator of the health status of the fish because they show an important role in the nonspecific or innate immune response. The higher the number of white blood cells (WBCs), the better an animal’s ability to perform well under stressful conditions [122]. Interestingly, the fish groups fed higher OLE concentrations (0.2% and 0.3%) exhibited higher basophil counts either pre- or post-infection compared to the control group. Roitt et al. [123] stated that after initiating a proper stimulus due to antigen injection, basophils count markedly changed in response to antigen injection. Similarly, Baba et al. [26] demonstrated allergic reactions as side effects of herbals due to their constituents in excessive doses.

Blood biochemical and serum metabolites could be utilized to recognize the infectious diseases [13]. The aberrant serum biochemical findings and the inflammatory progression to liver injury are the major features of an infection with *A. hydrophila* [124,125]. In this study, the *A. hydrophila* infection caused a significant increase in the serum Ast and Alt activities, BUN, and creatinine levels with reduced concentrations of total proteins and albumin compared to control the group supported by the histopathological changes in hepatopancreas in the form of vacuolar degeneration, hemosiderosis, and congestion of hepatic and pancreatic blood vessels together with raised mRNA transcriptional levels of hepatic *tnfα*, *il1β*, *caspase-3,* and *keap1* with inhibited *sod*, *nrf2*, and *pkc* expression levels referring to oxidative stress and inflammation. Moreover, the 0.1% OLE succeeded to modulate these adverse findings either before or after the bacterial challenge. Baba et al. [26] and Zemheri-Navruz et al. [33] reported improved serum biochemical parameters in the rainbow trout treated with the 0.1% OLE, while the high concentration of the OLE suppressed the immune function of the common carp.

The outcomes for the metabolic items used in the current study revealed that the serum triglycerides and cholesterol reduction occurred in response to the augmented influx of the fatty acids from the adipose tissue to the liver [126]. Notably, in mice fed high-fat diet, the OLE significantly decreased the visceral fat pad weight and plasma levels of both triglyceride and free fatty acids. This may have resulted in lower plasma levels of FFA, which flow into the liver and may result in less triglyceride synthesis there [98]. 

Concerning the nonspecific immune response, as the first line of defense system, is especially important to protect fish against invading pathogens [126]. The results of our study revealed that long-term administration of the OLE improved the immune response (WBCS, phagocytic and lysozyme activities, NBT, and total proteins) in addition to IgM and globulins concentrations with the best cumulative survival rate after the *A. hydrophila* challenge, while the opposite of these findings was detected in the infected unsupplied fish group. The increase in lysozyme in the blood of the stimulated fish is associated either with the proliferating phagocytes (monocytes and heterophils) or the increased amounts of lysozymes produced from the lysosomes [126]. Lysozyme activity is one of the top indicators for assessing the bactericidal impact of feed additives. Additionally, the flavonoids in the OLE are thought to be one of the most important stimulators of the innate immune system that limit microbial adhesion and colonization, may be the cause of the increased lysozyme activity [127]. Hoseinifar et al. [128] found a significant effect on serum lysozyme activity of *O. mykiss* fed the OLE. Baba et al. [26] also showed that the 0.1% of OLE can control Yersinia ruckeri infection effectively due to the ability of the OLE to enhance the immune system of the rainbow trout. Fish heterophils contain various phagocytic, bactericidal, and respiratory burst (NBT) activities as a potent oxygen-dependent killing mechanism in phagocytic cells, such as monocytes/macrophages and neutrophils, and is regarded as a highly efficient nonspecific cellular defense mechanism necessary for the assessment of the general health of fish [129]. Here, we detected that the OLE appreciably increased the blood NBT post *A. hydrophila* infection, via increasing the oxidation levels in phagocytes, as a crucial factor in the general defense mechanisms in the fish to limit the spread of diseases [130]. 

Serum total proteins are influenced by serum immunoglobulins (e.g., IgM) as an indicator for the enhanced immune system of fish [131]. Enhanced blood IgM in the current study is in line with the increased lysozyme activity, suggesting immunomodulatory effects of the OLE on the *C. carpio* after the *A. hydrophila* challenge [128]. 

A crucial response to the oxidative stress triggered by the pathogen entry and respiratory burst activity is the production of antioxidants since this stress causes lipid peroxidation, which in turn affects fish health [13]. Numerous cellular defense mechanisms, such as Sod, Cat, and Gpx activities, can reflect the antioxidant capacity of aquatic species, which counteracts the harmful effects of excessive ROS [132]. Olive by-products have antioxidant and free-radical-scavenging characteristics, which are likely due to the significant amounts of phenolic compounds present in them [133].

In the present study, the 0.1% OLE inclusion exhibited a hepatoprotective role through a notable reduction of serum hepatic injury biomarkers Alt and Ast, a remarkable raise in the hepatic antioxidant activities of Sod, Cat, and Gpx with suppressed lipid peroxidation (hepatic MDA) in addition to the enhanced hepatic mRNA expression of *nrf2*, *pkc*, and *sod* either before or after the *A. hydrophila* challenge. The authors attributed that to the OLE secoiridoids, especially oleuropein and its derivatives, strong antioxidant compounds [134] which protect cells against prooxidants [90]. This may also be due to the radical-scavenging effects of the OLE, as supported by Abdel-Razek et al. [135], who observed that the antioxidant activity displayed that the DPPH-radical-scavenging ability for aqueous extracts of olive leaves was 66.49%. 

For a more in-depth interpretation, the body has an effective defense system that detoxifies, eliminates toxic substances, and inactivates ROS to deal with oxidative damage. The essential redox-sensitive transcription factor *nrf2*, which serves as cells’ principal defense against the oxidative stress, is inhibited by DNA damage in infected cells [136]. The current study revealed that following the *A. hydrophila* challenge, the 0.1% OLE upregulated antioxidant-related genes (*sod*, *nrf2*, and *pkc*) while downregulating the immune-related genes (*tnfα*, *il1β,* and *keap1*) and apoptotic gene (*caspase-3*) of the *C. carpio* five days after the bacterial challenge compared to the other treatment groups with higher concentrations, as portrayed in Figure 8. The *Nrf2* activators lower the ROS levels and prevent ROS from activating *nf-kb-* and NF-kB-dependent inflammatory mediators (such as *il1β*, *il-1*, *il-6*, *tnf*, and *cox-2*) through ROS [137]. By preventing the transcription of pro-inflammatory cytokines, *nrf2* reduces inflammatory reactions [138]. Previous studies have shown that the OLE can reduce the release of pro-inflammatory substances including Il-1b*il-1* and *tnf* by the stomach [139]. Bedouhene et al. [140] and Bucciantini et al. [141] explained the powerful anti-inflammatory and antioxidant impact by preventing the degranulation of neutrophils, limiting the generation and release of inflammatory chemicals, and reducing the creation of ROS.

Additionally, in nonstressed conditions, *keap1* promotes *nrf2* breakdown, but oxidative insults directly alter the *keap1* thiol groups, inactivating *keap1* activity, stabilizing *nrf2*, and inducing cytoprotective genes. The production of ROS by stressed cells will result in the separation of the *nrf2-keap1* complex. When that happens, *nrf2* will act to activate the nucleus’s transcription of several genes involved in the antioxidant defenses and redox homeostasis [132,142]. While doing so, it controls cellular redox homeostasis, detoxification, glutathione homeostasis, and mitochondrial biogenesis. It does this by binding to the antioxidant response element in the promoter regions of several downstream antioxidation and detoxification enzymes, including Nqo1 (NAD(P)H dehydrogenase, quinone 1), Ho-1 (heme oxygenase-1), Sod, Catalase (Cat), and glutamate–cysteine-ligase catalytic subunit (GCLC) [143,144,145]. Whether or not the *Nrf2-Keap1* pathway appears to be dependent on the immune-cell type-, the immune-modulatory effect of *nrf2* influences the *nrf2-keap1* pathway and may protect host cells from a range of inflammatory illnesses. Studies in a septic mouse model, for instance, demonstrate that higher *nrf2* expression in *keap1-*/*M1* macrophages reduces the production of pro-inflammatory genes and ensures tissue harm [138]. From a different angle, disruption of *keap1* in murine myeloid leukocytes boosts peritoneal macrophages’ bacterial phagocytic activity [146]. Additionally, peritoneal neutrophils with Nrf2-deficiency (Nrf2/--) exhibit elevated levels of *tnf*-, *il-6* monocyte chemoattractant protein-1, and macrophage inflammatory protein-2 [147]. 

Compared to the fish fed the diets without the OLE, the 0.1% OLE-fed fish demonstrated tolerance to the *A. hydrophila* challenge, leading to significantly lower mortalities. The positive effects of the OLE on immunological function and antioxidant capacity are responsible for the higher survival rates in this study. It was claimed by Jiang et al. [148] that since the *A. hydrophila* pathogen and oxidative stress are closely related, the protective effect of the OLE may be related to the abundance of flavonoids and polyphenols in the OLE [26,33].

The study of pathological changes caused by various chemicals or biological infectious agents as biological markers is commonly conducted using histopathology [149]. Although fish immunity is influenced by the inclusion of immune cells in their tissues, the head, kidney, spleen, and liver are also involved [150]. In the current investigation, feeding the 0.1% OLE to the fish significantly reduced the pathological alterations brought on by *A*. *hydrophila* infection in all the studied organs. Hepatic expression of *il1β*, *tnfα*, and *caspase-3* levels were also downregulated compared to higher doses. In the same context, Jiang et al. [151] reported that upregulations [152] showed the increase in the hepatic expression levels of cytochrome-c and *caspase-3* in rats fed a high-fat diet for 12 to 16 weeks. Similar to the current results, other studies reported the antiapoptotic effects of olive leaf phenolics. It has been shown that oleuropein- and hydroxytyrosol-rich OLE reduces liver apoptosis and the disruption of lipid metabolism in the high-fat-diet-fed rats [152]. Furthermore, Alhaithloul et al. [153] illustrated that olive oil biophenols, which have direct antioxidant properties, prevented cyclophosphamide-induced oxidative stress and apoptosis in the rat kidney. Additionally, the OLE modulated intestinal epithelial homeostasis by favorably influencing inflammation and gut microbiota [154]. The extra virgin olive oil’s unsaponifiable fraction (UF) demonstrated anti-inflammatory and antioxidant properties by reducing the generation of intracellular ROS and nitrites brought on by Lipopolysaccharides [155]. Additionally, UF reduced the NF-kB-signal pathway and Mapk (mitogen-activated protein kinase) phosphorylation, which in turn inhibited the production of Cox-2 and *iNos* proteins [155].

## 5. Conclusions

Shreds of evidence on the ways by which the OLE and its polyphenols, such as oleuropein, could exert their growth-promoting, antioxidative, anti-inflammatory, and immunomodulatory properties targeting the Nrf2/kaep1 signaling pathway are provided by the current investigation supported by characterization of the OLE components. Our findings indicate that the OLE, when used in the *C. carpio* diets at a level of 0.1%, has a high capacity for growth performance by boosting nutrient utilization, whereas higher doses of the OLE must be avoided, since they may induce oxidative stress. As a result, the OLE at a dose of 0.1% is considered an organic natural product without adverse effects on the environment and fish as a safe and sustainable alternative to chemical growth promoters in aquaculture practice. This is manifested not only in increasing the antioxidant system activity in the cells but also in blocking the expression of the proteins involved in the inflammation caused by *A. hydrophila* infection.

## Figures and Tables

**Figure 1 animals-13-02229-f001:**
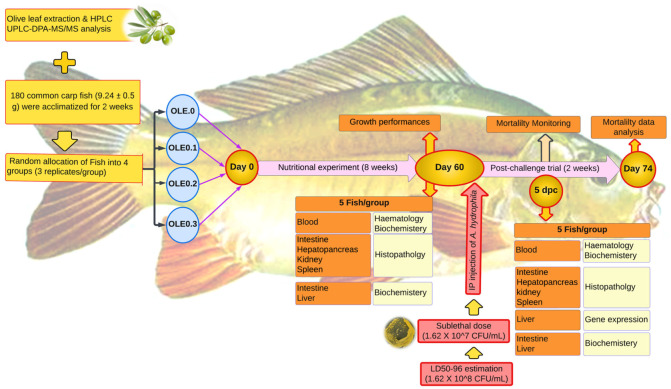
Experimental design.

**Figure 2 animals-13-02229-f002:**
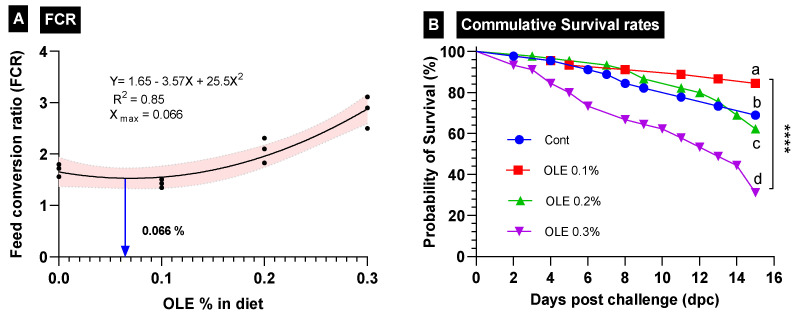
Growth performance parameters in different treatment groups of *C. carpio* fed OLE extract. (**A**) Feed conversion ratio (FCR) and (**B**) cumulative survival rates of all treatment groups. **** Different superscription letters indicate significant differences between groups.

**Figure 3 animals-13-02229-f003:**
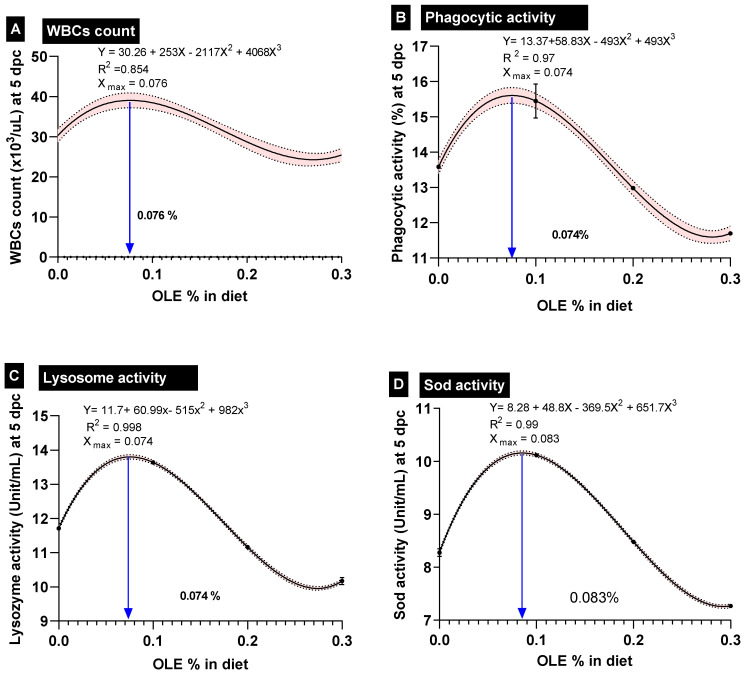
Polynomial-regression-based interpolation of the optimum-performing doses of olive leaf extract in common carp feed is given on each interpolation curve; dpc; days post-challenge. (**A**) WBCs: white blood cells, (**B**) phagocytic activity, (**C**) lysozyme activity, and (**D**) SOD: superoxide dismutase activity.

**Figure 4 animals-13-02229-f004:**
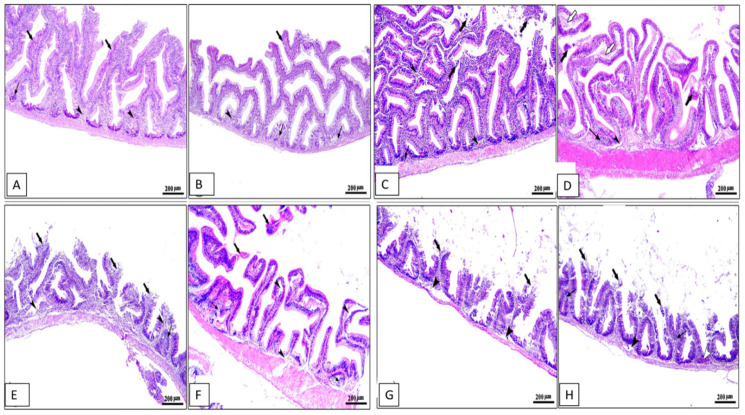
Photomicrograph of intestine sections of *Cyprinus carpio* stained with hematoxylin and eosin (H&E). (**A**–**D**) Pre-infected fish with *A. hydrophila* fed the control diet, 0.1%, 0.2%, and 0.3% OLE, respectively, showing intestinal villi with increased length and branching on plate (**B**) and degeneration and sloughing of apical part of villi on plates (**C**,**D**) with lamina propria of loose CT containing lymphocytic aggregations on plates (**B**–**D**) in addition to edema and degenerative changes in the lamina propria (white arrows) on plate (**D**). (**E**–**H**) Post-infected fish with *A. hydrophila* fed the control diet, 0.1%, 0.2%, and 0.3% OLE, respectively, showing severe degenerative changes in the intestinal villi with sloughing of lining epithelium into the intestinal lumen (black thick arrows), edema in the lamina propria (black arrow heads), and mononuclear cells infiltration (black thin arrows) besides the presence of hemorrhage (arrow heads).

**Figure 5 animals-13-02229-f005:**
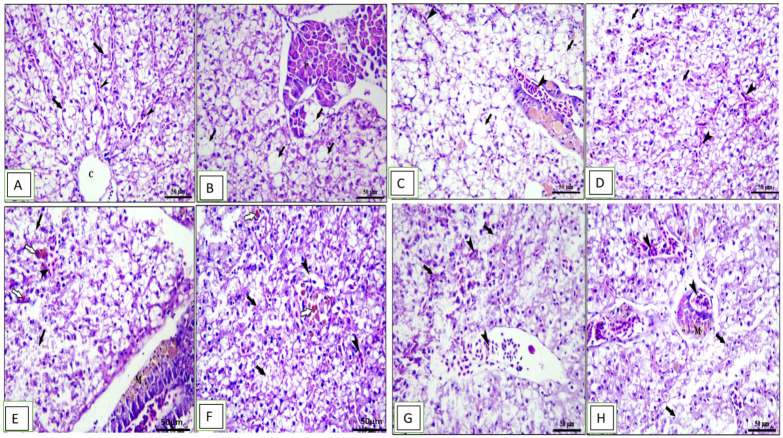
Photomicrograph of hepatopancreas sections of *Cyprinus carpio* stained with hematoxylin and eosin (H&E). (**A**–**D**) Pre-infected fish with *A. hydrophila* fed the control diet, 0.1%, 0.2%, and 0.3% OLE, respectively. (**A**) Normal architecture of liver with intact central vein “c” and polyhedral-shaped hepatocytes (arrows) separated by blood sinusoids (arrow heads). (**B**) Mild vacuolar changes in hepatocytes and pancreatic acini (arrows). (**C**) Moderate vacuolar changes in hepatocytes (arrows) and congestion in both hepatic sinusoids and pancreatic blood vessels (arrow heads). (**D**) Severe vacuolar degeneration in hepatocytes (arrows) and congestion in hepatic sinusoids (arrow heads). (**E**–**H**) Post-infected fish with *A. hydrophila* fed the control diet, 0.1%, 0.2%, and 0.3% OLE, respectively. (**E**) Nuclear pyknosis, vacuolar degeneration (black arrows), hemosiderosis (white arrows), and congestion of hepatic and pancreatic blood vessels (arrow heads) in addition to the presence of melanomacrophage centers (m).

**Figure 6 animals-13-02229-f006:**
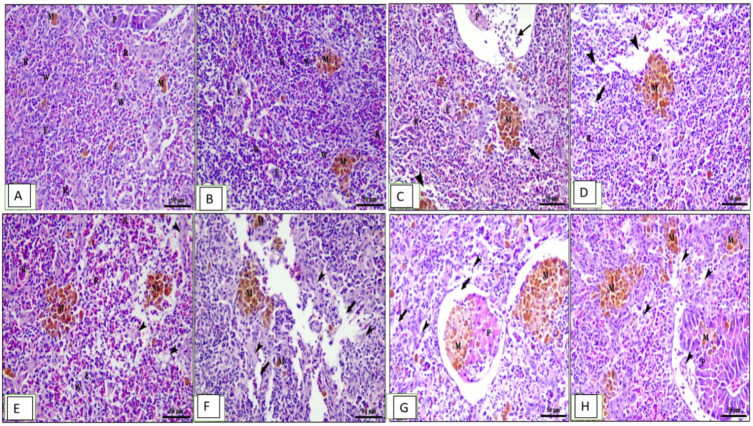
Photomicrograph of spleen sections of *Cyprinus carpio* stained with hematoxylin and eosin (H&E). (**A**–**D**) Pre-infected fish with *A. hydrophila* fed the control diet, 0.1%, 0.2%, and 0.3% OLE, respectively.(**A**) Mixed white (w) and red pulp (p), ellipsoids (**e**), melanomacrophage centers (M), and pancreatic acini (p). (**B**) Congestion of splenic blood vessels (thin arrows), degenerative changes, and lymphocyte depletion (thick arrows) besides interstitial edema (arrow heads). (**C**,**D**) The severity of the lesion increased in a dose–response relationship. (**E**–**H**) Post-infected fish with *A. hydrophila* fed the control diet, 0.1%, 0.2%, and 0.3% OLE, respectively. (**E**) Mixed white (w) and red pulp (p), ellipsoids (e)], melanomacrophage centers (m)], degeneration of pancreatic acinar cells (p), and degenerative changes and lymphocyte depletion (arrow heads) besides interstitial edema (arrows).

**Figure 7 animals-13-02229-f007:**
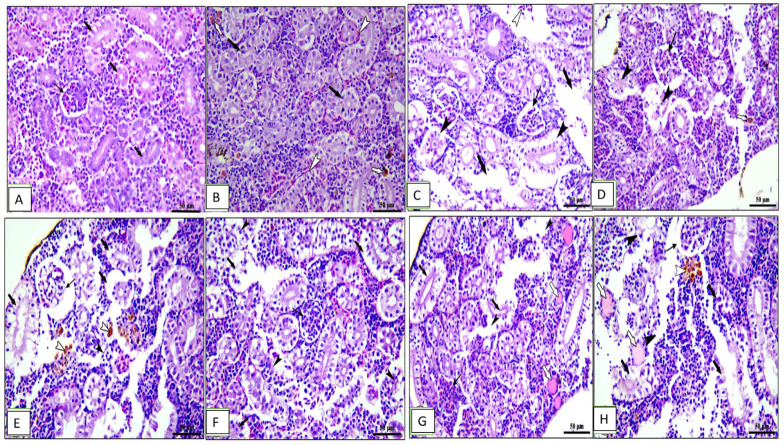
Photomicrograph of kidney sections of *Cyprinus carpio* stained with hematoxylin and eosin (H&E). (**A**–**D**) Pre-infected fish with *A. hydrophila* fed the control diet, 0.1%, 0.2%, and 0.3% OLE, respectively. (**A**) Intact renal glomeruli. (**B**) Intact renal tubules on plates (**A**,**B**) (thick black arrows). (**C**) Degeneration (thin black arrows) and sloughing of tubular epithelium (black arrow heads), congestion of blood vessels on plates (**B**,**C**) (white arrow heads), hemosiderosis on plate (**D**) (0.3% OLE) (white arrows). (**E**–**H**) Post-infected fish with *A. hydrophila* fed the control diet, 0.1%, 0.2%, and 0.3% OLE, respectively. (**E**) Degeneration in renal glomeruli (thin black arrows), degeneration and separation of tubular epithelium (thick black arrows), presence of hyaline cast (white arrows), and edema and degeneration of interstitial tissue (black arrow heads) in addition to hemosiderosis (white arrow heads).

**Figure 8 animals-13-02229-f008:**
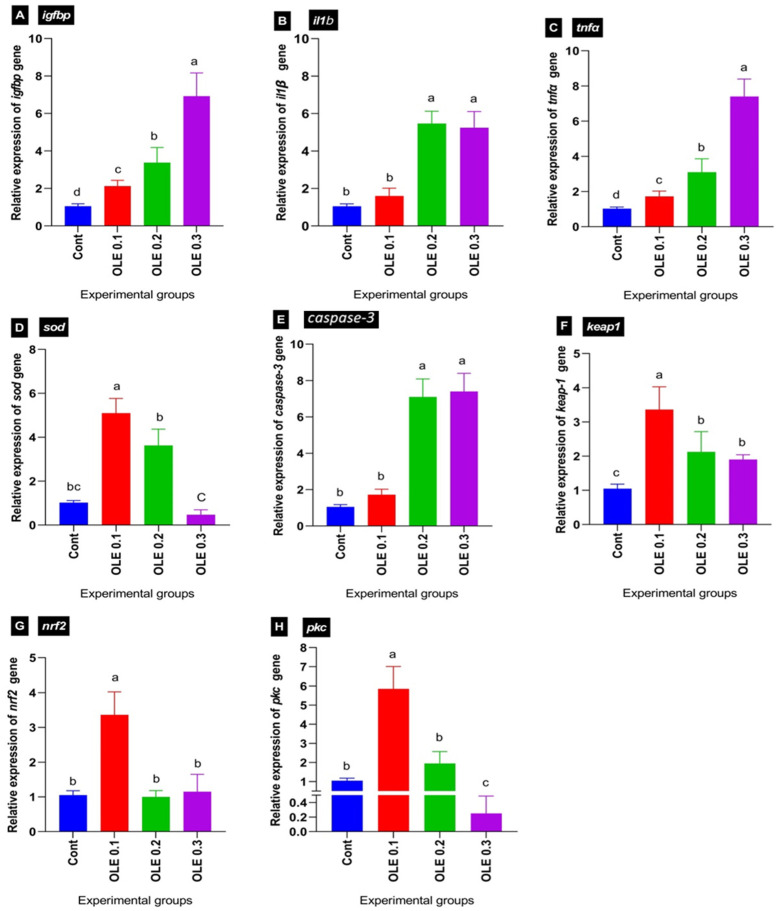
Differential expression of different genes in common carp groups fed OLE then challenged with *Aeromonas hydrophila.* (**A**) Insulin-like growth factor-binding proteins: *igfbp*. (**B**) Interleukin 1β: *il1β*. (**C**) Tumor necrosis factor α: *tnfα*. (**D**) Superoxide dismutase: *sod*. (**E**) *Caspase-3*. (**F**) Kelch-like ECH-associated protein 1: *keap1*. (**G**) Nuclear factor erythroid 2-related factor 2: *nrf2*. (**H**) Protein kinase C: *pkc*. Columns with different superscript letters in the same figure are significantly different (*p* ≤ 0.05).

**Table 1 animals-13-02229-t001:** Ingredient composition and proximate analysis percentages as-is of the experimental diets designed to contain graded levels of olive leaf extract (OLE) fed to juvenile *Cyprinus carpio* for 60 days.

	Experimental Diets			
	Control	Diet 1	Diet 2	Diet 3
	0% OLE	0.1% OLE	0.2% OLE	0.3% OLE
Ingredients%				
Fish meal (65% Cp)	10.00	10.00	10.00	10.00
Soybean meal (45% Cp)	39.00	39.00	39.00	39.00
Corn gluten meal	10.30	10.30	10.30	10.30
Yellow corn	10.00	10.00	10.00	10.00
Wheat flour	25.80	25.70	25.60	25.50
Soybean oil	3.70	3.70	3.70	3.70
Vitamin premix *	0.15	0.15	0.15	0.15
Mineral premix **	0.15	0.15	0.15	0.15
Dicalcium phosphate	0.60	0.60	0.60	0.60
Choline chloride	0.20	0.20	0.20	0.20
Stay C ***	0.10	0.10	0.10	0.10
Olive leaf extract	0.00	0.10	0.20	0.30
Composition (%)				
Crude protein	33.61	33.60	33.59	33.58
DE (Kcal/Kg)	2999.78	2998.19	2996.60	2995.02
Crude lipid	5.90	5.90	5.90	5.90
Ash	4.79	4.79	4.79	4.79
Crude fiber	3.61	3.61	3.61	3.61
Ca	0.78	0.78	0.78	0.78
P	0.82	0.82	0.81	0.81

* Vitamin (g/kg premix): thiamin HCl, 0.44; riboflavin, 0.63; pyridoxine HCl, 0.91; DL-pantothenic acid, 1.72; nicotinic acid, 4.58; biotin, 0.21; folic acid, 0.55; inositol, 21.05; menadione sodium bisulfite, 0.89; vitamin A acetate, 0.68; vitamin D3, 0.12; DL-alpha-tocopherol acetate, 12.63; alpha-cellulose, 955.59. ** Trace mineral (g/100 g premix): cobalt chloride, 0.004; cupric sulfate pentahydrate, 0.25; ferrous sulfate, 4.000; magnesium sulfate anhydrous, 13.862; manganous sulfate monohydrate, 0.650; potassium iodide, 0.067; sodium selenite, 0.010; zinc sulfate heptahydrate, 13.193; alpha-cellulose, 67.964. *** Stay C^®^, (L-ascorbyl-2-polyphosphate 35%).

**Table 2 animals-13-02229-t002:** Primer sequences for *C. carpio* genes used in this study.

Gene ID	Primer Sequence (5′–3′)	NCBI Gene Bank Acc. Number	Reference
** *sod* **	F: GGCTTTGATAAGGACAGTGGAA GACT R: GAAGTGGGACGAGACCTGTAGTG	AJ492825	[70]
** *il1β* **	F: ACCGGCACACGTTACAACACTT R: GGGTGGTTGGCATCTGGTTCAT	AJ245635.1	[71]
** *Tnfα* **	F: AACCAGGACCAGGCTTTCACT3 R: CATGTAGCGGCCATAGGAATC3	AJ311800,2	[71]
** *igfbp* **	F: CAAAGGCAACGCAATACGC R: GACCGTGTTTGTCACAGTTTGGA	MG919989	[72]
***β-actin* ***	F: ATCCGTAAAGACCTGTATGCCA R: GGGGAGCAATGATCTTGATCTTC	JQ619774	[70]
** *caspase* *-3* **	F: CTCTACGGCACCAGGTTACTACTC R: GCCATCATTTCACAAAGGGACT	KF055462	[73]
** *pkc* **	F: TGGGCGTCCTGATGTTTGAG R: GGCGTTCCTTTGGTTCCTTG	JX673919	[73]
** *nrf2* **	F: TTCCCGCTGGTTTACCTTAC R: CGTTTCTTCTGCTTGTCTTT	XM_019123954.1	[74]
** *keap1* **	F: GCTCTTCGGAAACCCCT R: GCCCCAAGCCCACTACA	XM_019071157.1	

* ***beta-actin***: internal reference gene.

**Table 3 animals-13-02229-t003:** Growth and feed utilization parameters of common carp fed different doses of OLE.

	Control	OLE 0.1%	OLE 0.2%	OLE 0.3%	*p*-Value
IBW (g)	9.24 ± 0.19	9.12 ± 0.17	9.18 ± 0.14	9.26 ± 0.21	0.841
FBW (g)	15.72 ± 0.17 ^b^	16.48 ± 0.23 ^a^	14.11 ± 0.26 ^c^	12.60 ± 0.33 ^d^	<0.0001
WG (%)	71.4 ± 2.1 ^b^	80.76 ± 2.77 ^a^	55.2 ± 3.01 ^c^	40.7 ± 3.12 ^d^	<0.0001
FCR	1.64 ± 0.08 ^c^	1.43 ± 0.07 ^d^	2.08 ± 0.11 ^b^	2.78 ± 0.13 ^a^	<0.0001
SGR (%/day)	0.85 ± 0.03 ^ab^	0.97 ± 0.04 ^a^	0.72 ± 0.05 ^b^	0.53 ± 0.02 ^c^	0.0003

IBW: initial body weight; FBW: final body weight; WG: weight gain; FCR: feed conversion ratio; SGR: specific growth rate; PER: protein efficiency rate. Data are expressed as mean ± SEM, where *n* = triplicate tanks for WG%, FCR, and SGR. Values with different superscripts within a row are significantly different (*p* < 0.05).

**Table 4 animals-13-02229-t004:** Hematological findings of the control and OLE-treated groups before and after *A. hyrophila* infection.

		Control	OLE 0.1%	OLE 0.2%	OLE 0.3%	*p*-Value of Two-Way ANOVA *		
	Time to Challenge					OLE Dose	Infection	Interaction
RBCs (×10^6^ µL^−1^)	Pre	1.81 ± 0.11	1.83 ± 0.08	1.79 ± 0.14	1.84 ± 0.09	0.846	0.235	0.928
	Post	1.92 ± 0.12	1.93 ± 0.17	1.89 ± 0.16	1.91 ± 0.11			
Hb (g dL^−1^)	Pre	6.41 ± 0.16	6.72 ± 0.19	6.53 ± 0.09	6.69 ± 0.14	0.714	0.047	0.1108
	Post	6.87 ± 0.12	6.56 ± 0.19	7.08 ± 0.17	6.74 ± 0.21			
PCV (%)	Pre	23.09 ± 2.2	25.23 ± 1.9	24.21 ± 2.7	23.87 ± 1.9	0.330	0.203	0.360
	Post	26.11 ± 2.34	25.85 ± 2.6	26.13 ± 3.2	26.78 ± 2.5			
MCV (fL)	Pre	127.6 ± 7.45 ^c^	137.9 ± 8.43 ^ab^	135.3 ± 6.86 ^ab^	129.73 ^bc^	0.001	0.053	0.0180
	Post	136.0 ± 4.5 ^abc^	134.9 ± 5.2 ^abc^	138.3 ± 3.1 ^a^	140.21 ± 3.43 ^a^			
MCH (pg)	Pre	35.41 ± 3.21	36.72 ± 2.1	35.8 ± 3.4	36.34 ± 2.8	0.2385	0.3163	0.1407
	Post	35.78 ± 1.8	34.09 ± 1.4	37.46 ± 1.09	35.29 ± 1.12			
MCHC (%)	Pre	27.80 ± 1.12 ^ab^	26.63 ± 1.2 ^ab^	26.97 ± 1.31 ^ab^	28.03 ± 1.42 ^a^	0.2385	0.00163	0.01407
	Post	25.97 ± 1.31 ^b^	25.38b ± 1.01 ^b^	27.1 ± 1.91 ^ab^	27.2 ± 1.99 ^ab^			
WBCs (×10^3^ µL^−1^)	Pre	22.11 ± 1.12 ^d^	25.6 ± 1.32 ^c^	23.89 ± 1.08 ^cd^	24.38 ± 1.2 ^cd^	<0.0001	<0.0001	<0.0001
	Post	28.26 ± 1.09 ^b^	31.45 ± 1.12 ^b^	33.21 ± 1.3 ^a^	35.42 ± 1.1 ^a^			
Lymphocytes (×10^3^ µL^−1^)	Pre	18.34 ± 1.6 ^c^	21.71 ± 1.32 ^b^	18.9 ± 0.94 ^c^	20.32 ± 1.11 ^b^	<0.0001	<0.0001	0.0002
	Post	20.89 ± 0.93 ^b^	25.08 ± 1.1 ^a^	26.2 ± 1.43 ^a^	27.99 ± 0.94 ^a^			
Monocytes (×10^3^ µL^−1^)	Pre	0.71 ± 0.08 ^c^	0.81 ± 0.1 ^c^	1.67 ± 0.09 ^b^	0.75 ± 0.06 ^c^	<0.0001	<0.0001	<0.0001
	Post	1.49 ± 0.1 ^b^	2.75 ± 0.09 ^a^	2.81 ± 0.07 ^a^	2.74 ± 0.04 ^a^			
Heterophils (×10^3^ µL^−1^)	Pre	2.48 ± 0.06 ^b^	2.54 ± 0.05 ^b^	2.47 ± 0.08 ^b^	2.40 ± 0.1 ^b^	<0.45	<0.0001	<0.0001
	Post	4.99 ± 0.1 ^a^	2.93 ± 0.1 ^b^	2.69 ± 0.08 ^b^	3.98 ± 0.07 ^a^			
Basophils (×10^3^ µL^−1^)	Pre	0.58 ± 0.12 ^c^	0.54 ± 0.09 ^c^	0.85 ± 0.13 ^b^	0.91 ± 0.11 ^b^	<0.0001	0.7445	0.0001
	Post	0.89 ± 0.12 ^b^	0.69 ± 0.1 ^bc^	1.51 ± 0.11 ^a^	1.71 ± 0.15 ^a^			

Data are expressed as mean ± SEM, where *n* = 5/replicate. * Two-way ANOVA revealed significant effect of the OLE dose or infection, one-way ANOVA was run to demonstrate the pairwise comparison between experimental groups within the same time point, while in parameters with significant interaction, all treatments from nonchallenged groups were compared to the challenged groups, using one-way ANOVA followed by post hock multiple comparison. Superscription letters indicate the significant differences between experimental groups. RBCs: red blood corpuscles; Hb: hemoglobin; PCV: packed cell volume; MCV: mean corpuscular volume; MCH: mean corpuscular hemoglobin; MCHC: mean corpuscular hemoglobin concentration; WBCs: white blood cells.

**Table 5 animals-13-02229-t005:** Serum biochemical, intestinal digestive and ALK, and hepatic oxidative stress and antioxidant biomarkers of the control and OLE-fed groups pre- and post-challenge.

Parameter	Pre- and Post-Challenge	Control	OLE 0.1%	OLE 0.2%	OLE 0.3%	*p*-Value		
						OLE Dose	Infection	Interaction
Serum biochemistry								
Total proteins (g dL^−1^)	Pre	4.64 ± 0.09 ^c^	5.04 ± 0.06 ^a^	4.76 ± 0.1 ^b^	5.05 ± 0.07 ^a^	<0.0001	0.074	0.0001
	Post	4.84 ± 0.03 ^b^	4.99 ± 0.05 ^a^	4.78 ± 0.04 ^b^	5.02 ± 0.1 ^a^			
Albumin (g dL^−1^)	Pre	1.70 ± 0.08 ^b^	1.79 ± 0.09 ^a^	1.65 ± 0.03 ^b^	1.59 ± 0.07 ^bc^	0.0001	0.0001	0.0001
	Post	1.67 ± 0.07 ^b^	1.63 ± 0.04 ^b^	1.56 ± 0.06 ^c^	1.40 ± 0.05 ^d^			
Globulins (g dL^−1^)	Pre	2.94 ± 0.05 ^d^	3.30 ± 0.1 ^bc^	3.11 ± 0.09 ^c d^	3.46 ± 0.1 ^b^	0.0001	0.0001	0.001
	Post	3.17 ± 0.1 ^c^	3.67 ± 0.07 ^a^	3.22 ± 0.08 ^c^	3.62 ± 0.1 ^a^			
Cholesterol (mg dL^−1^)	Pre	105.2 ± 4.2 ^a^	102.0 ± 3.4 ^a^	98.43 ± 2.5 ^b^	97.12 ± 1.8 ^b^	0.0002	0.0046	0.01
	Post	101.3 ± 2.9 ^ab^	98.54 ± 4.1 ^b^	94.68 ± 2.7 ^c^	92.08 ± 3.5 ^c^			
Triglycerides (mg dL^−1^)	Pre	104.30 ± 3.1 ^bc^	88.10 ± 3.5 ^d^	100.7 ± 2.8 ^c^	108.31 ± 1.6 ^b^	0.0001	0.009	0.0001
	Post	115.97 ± 3.6 ^a^	99.58 ± 2.5 ^c^	110.0 ± 2.8 ^b^	120.40 ± 2.3 ^a^			
AST (U L^−1^)	Pre	24.07 ± 1.2 ^d^	24.15 ± 1.6 ^d^	33.02 ± 1.1 ^c^	39.58 ± 0.8 ^b^	0.0001	0.0001	0.0001
	Post	39.71 ± 1.8 ^b^	28.92 ± 1.1 ^cd^	39.93 ± 1.7 ^b^	49.37 ± 2.1 ^a^			
ALT (U L^−1^)	Pre	30.20 ± 1.2 ^c^	29.35 ± 2.3 ^c^	39.72 ± 2.1 ^b^	42.56 ± 1.5 ^b^	0.001	0.001	0.006
	Post	49.10 ± 2.2 ^a^	38.54 ± 2.4 ^b^	49.95 ± 1.8 ^a^	52.36 ± 3.1 ^a^			
BUN (mg dL^−1^)	Pre	2.7 ± 0.1 ^c^	2.34 ± 0.09 ^d^	3.12 ± 0.1 ^b^	3.67 ± 0.1 ^a^	0.0001	0.0001	0.0001
	Post	2.9 ± 0.1 ^c^	2.45 ± 0.07 ^d^	3.0 ± 0.05 ^bc^	3.19 ± 0.1 ^b^			
Creatinine (mg dL^−1^)	Pre	0.37 ± 0.08 ^d^	0.36 ± 0.04 ^d^	0.40 ± 0.06 ^c^	0.49 ± 0.03 ^b^	0.0001	0.0001	0.001
	Post	0.45 ± 0.07 ^b^	0.40 ± 0.05 ^c^	0.43 ± 0.06 ^bc^	0.63 ± 0.08 ^a^			
Intestinal enzyme activity								
Protease (U mg^−1^)	Pre	48.20 ± 1.3 ^b^	52.00 ± 1.4 ^a^	50.03 ± 2.1 ^a^	44.34 ± 1.3 ^b^	0.0001	0.0001	0.332
	Post	40.67 ± 2.1 ^b^	45.27 ± 1.9 ^a^	40.97 ± 1.7 ^b^	38.08 ± 1.5 ^c^			
Amylase (U mg^−1^)	Pre	77.42 ± 3.2 ^a^	81.08 ± 2.6 ^a^	79.30 ± 2.1 ^a^	68.32 ± 1.9 ^b^	0.019	0.016	0.01
	Post	67.92 ± 2.7 ^b^	78.47 ± 2.1 ^a^	69.04 ± 1.7 ^b^	60.59 ± 2.1 ^c^			
Lipase (U mg^−1^)	Pre	65.90 ± 2.1 ^b^	74.80 ± 3.2 ^a^	60.00 ± 2.9 ^b^	58.60 ± 1.3 ^b^	0.001	0.001	0.001
	Post	62.10 ± 2.9 ^b^	68.68 ± 2.0 ^ab^	58.80 ± 2.2 ^b^	48.32 ± 2.4 ^c^			
Intestinal alkaline phosphatase (U mg^−1^)	Pre	73.12 ± 1.21 ^b^	82.65 ± 1.94 ^a^	45.11 ± 1.2 ^d^	44.99 ± 1.16 ^d^	0.0001	0.0001	0.014
	Post	54.95 ± 0.94 ^c^	73.34 ± 1.32 ^b^	58.05 ± 1.31 ^c^	53.16 ± 0.89 ^c^			
Hepatic antioxidant activities (U mg^−1^)								
SOD (U mg^−1^)	Pre	7.320 ± 0.2 ^b^	10.360 ± 0.13 ^a^	6.51 ± 0.1 ^b^	5.780 ± 0.12 ^c^	<0.0001	0.0032	0.0145
	Post	6.280 ± 0.11 ^b^	9.12 ± 0.10 ^a^	5.70 ± 0.13 ^c^	4.270 ± 0.14 ^d^			
GPx (U mg^−1^)	Pre	8.160 ± 0.2 ^b^	9.97 ± 0.09 ^a^	7.910 ± 0.08 ^b^	6.580 ± 0.09 ^c^	<0.0001	0.1552	0.5508
	Post	7.0 ± 0.11 ^b^	8.17 ± 0.12 ^a^	6.24 ± 0.08 ^b^	5.08 ± 0.1 ^c^			
CAT (U mg^−1^)	Pre	11.52 ± 0.45 ^a^	12.35 ± 0.32 ^a^	11.07 ± 0.4 ^b^	10.93 ± 0.7 ^b^	0.009	0.0174	0.0016
	Post	10.70 ± 0.4 ^c^	11.45 ± 0.3 ^a b^	10.18 ± 1.47	9.95 ± 0.4 ^c^			
MDA (nmol mg^−1^)	Pre	17.19 ± 0.19 ^b^	14.32 ± 0.12 ^c^	17.79 ± 0.16 ^a^	19.84 ± 0.15 ^a^	<0.0001	0.0053	0.1168
	Post	20.21 ± 0.13 ^b^	15.38 ± 0.51 ^c^	19.48 ± 0.43 ^b^	22.87 ± 0.35 ^a^			
Immune parameters								
Phagocytic activity%	Pre	12.09 ± 1.2 ^a^	12.5 ± 0.9 ^a^	11.00 ± 1.1 ^b^	10.18 ± 0.7 ^b^	<0.0001	<0.0001	0.1921
	Post	13.58 ± 1.9 ^b^	15.45 ± 1.3 ^a^	12.98 ± 1.7 ^b^	11.70 ± 1.3 ^c^			
Lysosome activity (U mL^−1^)	Pre	8.980 ± 0.11 ^b^	10.22 ± 0.19 ^a^	8.860 ± 0.1 ^b^	8.110 ± 0.12 ^b^	0.001	0.001	0.103
	Post	11.71 ± 0.1 ^b^	13.64 ± 0.13 ^a^	11.16 ± 0.1 ^b^	10.17 ± 0.08 ^c^			
NBT	Pre	0.20 ± 0.02 ^bc^	0.34 ± 0.02 ^a^	0.25 ± 0.02 ^b^	0.29 ± 0.02 ^b^	0.001	0.01	0.0213
	Post	0.15 ± 0.02 ^c^	0.39 ± 0.02 ^a^	0.29 ± 0.02 ^b^	0.32 ± 0.02 ^a^			
IgM (mg dL^−1^)	Pre	2.5 ± 0.05 ^c^	3.00 ± 0.1 ^b^	2.70 ± 0.09 ^c^	3.10 ± 0.1 ^b^	0.0001	0.01	0.016
	Post	3.07 ± 0.1 ^b^	3.37 ± 0.07 ^a^	3.20 ± 0.08 ^a^	3.080 ± 0.1 ^b^			

Data are expressed as mean ± SEM, where *n* = 5/replicate. Two-way ANOVA revealed significant effect of the OLE dose or infection, one-way ANOVA was run to demonstrate the pairwise comparison between experimental groups within the same time point, while in parameters with significant interaction, all treatments from nonchallenged groups were compared to the challenged groups, using one-way ANOVA followed by post hock multiple comparison. Superscription letters indicate the significant differences between experimental groups. SOD: superoxide dismutase; GPX: glutathione peroxidase; CAT: catalase; MDA: malondialdehyde; NBT: nitroblue tetrazolium test; IgM: immunoglobulin M. Different letters indicate significantly different means.

## Data Availability

The authors confirm that the data supporting the findings of this study are available within the article and/or its Appendix A.

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
