# Peer review of "Dietary Olive Leaf Extract Differentially Modulates Antioxidant Defense of Normal and Aeromonas hydrophila-Infected Common Carp (Cyprinus carpio) via Keap1/Nrf2 Pathway Signaling: A Phytochemical and Biological Link"

_animals, 2023, doi:10.3390/ani13132229_

Round 1

Reviewer 1 Report

The article is interesting, dealing with the inclusion an olive oil component in the diet of carp that, after a period, were challenged with A. hydrophila. Although the work is incremental, it was designed and carried out with the necessary scientific rigor and the results relevant to science and the breeders of this species. There are a few points I would like clarified:

- mg/L, CFU/L and other units must be changed to mg L-1, CFU L-1 etc.

- The summary states that in the hematological analyzes the WBC was analyzed, however in the material and methods and results there are other analyzes such as RBC, PVG, etc. Add to summary.

- In figures 5, 6, 7 and 8, improve the quality of the indications as they are difficult to visualize. Subtitles should also be reviewed and improved.

- Page 6, line 233 - in item 2.5 - Challenge with A. hydrophila, in the first line the phrase ..... fish from diet groups were inoculated with PCR-identified pathogenic Aeromonas hydrophila..... are control fish also included here?

- Table 5 - Hetrophils (I believe it is Heterophils). Here, the most common nomenclature for fish is neutrophils, since heterophiles are rarely found in this group of animals. No eosinophils were found? In the discussion it is said that the challenge provoked a certain allergic reaction (page 24, line 807).

-Page 24, line 758- the sentence "In agreement with Sudjana et al. (106); Gullón et al. (107); Centrone et al. (108) who reported antibacterial efficacy against Staphylococcus aureus and Escherichia coli". it's meaningless. Loose phrase.

Page 24, lines 801 and 802 - Review the sentence: The more the elevated numbers of WBCs the healthier the capacity of an animal to perform well underneath stressful conditions and fight diseases (123).

Page 24, line 832: ..... the immune response (WBCS, phagocytic and.....

would it be .... the immune response (WBCS), phagocytic and....?

Moderate editing of English language required

Author Response

Dear respected reviewer, we thank you for your time and valuable comments and appreciate your effort to improve our manuscript. Here we mentioned point-by-point response to the reviewers’ comments in red colour and highlighted in the manuscript with yellow colour.

  • mg/L, CFU/L and other units must be changed to mg L-1, CFU L-1 etc.

Changed

- The summary states that in the hematological analyzes the WBC was analyzed, however in the material and methods and results there are other analyzes such as RBC, PVG, etc. Add to summary.

Added

- In figures 5, 6, 7 and 8, improve the quality of the indications as they are difficult to visualize. Subtitles should also be reviewed and improved.

Done

- Page 6, line 233 - in item 2.5 - Challenge with A. hydrophila, in the first line the phrase ..... fish from diet groups were inoculated with PCR-identified pathogenic Aeromonas hydrophila..... are control fish also included here?

Yes the control group was included in the challenge and it was mentioned in the experimental design figure. Also corrected in this part.

- Table 5 - Hetrophils (I believe it is Heterophils). Here, the most common nomenclature for fish is neutrophils, since heterophiles are rarely found in this group of animals. No eosinophils were found? In the discussion it is said that the challenge provoked a certain allergic reaction (page 24, line 807).

Corrected

-Page 24, line 758- the sentence "In agreement with Sudjana et al. (106); Gullón et al. (107); Centrone et al. (108) who reported antibacterial efficacy against Staphylococcus aureus and Escherichia coli". it's meaningless. Loose phrase.

Corrected

Page 24, lines 801 and 802 - Review the sentence: The more the elevated numbers of WBCs the healthier the capacity of an animal to perform well underneath stressful conditions and fight diseases (123).

Done

Page 24, line 832: ..... the immune response (WBCS, phagocytic and.....

would it be .... the immune response (WBCS), phagocytic and....?

These parameters are all indicators of the immune response of the animal that is why they were all included in brackets and not separated.

Reviewer 2 Report

Very comprehensive, wide-ranging and scientifically robust work, presented in a correct and rigorous manner about the impact of OLE on Cyprinus carpio biology with relevant implications for its rearing.

Only minor issues /suggestions / recommendations:

- line 113, ... cultured freshwater fish species.... ;

- line 118, first time A. hydrophila is named in the main text should be written without abbreviation;

- line 206, do not forget to put plus and minus symbol por DO, pH and EC;

- Table 1, only one collum is sufficient to present diet composition since the only variation factor is the olive leaf extract;

- Figure 1, random instead of "randum" replicate instead of "replica" hydrophila instead of "hidrophia";

- line 235, explain how do you estimated LD50-96;

- line 284, may the first full stop in the sentence is not corrected;

- Figure 2, Figure 3 Figure 4, please say in the legend what are the cases named with capital letters (A; B, C, D,...)

- Line 510, A. hydrophila

- Line 514, take out the text filler;

- Figure 5, say in the legend what the cases E, F, G, H means;

- Figure 6, say in the legend what the cases F, G, H means;

- At the end of the discussion I would like to see what do you suggest for further research. For instance, do you think it is feasible to isolate and test only the considered positive substances in carp diet?

Author Response

Dear respected reviewer, we thank you for your time and valuable comments and appreciate your effort to improve our manuscript. Here we mentioned point-by-point response to the reviewers’ comments in red colour and highlighted in the manuscript with yellow colour.

Very comprehensive, wide-ranging and scientifically robust work, presented in a correct and rigorous manner about the impact of OLE on Cyprinus carpio biology with relevant implications for its rearing.

Only minor issues /suggestions / recommendations:

- line 113, ... cultured freshwater fish species.... ;

Done

- line 118, first time A. hydrophila is named in the main text should be written without abbreviation;

Corrected

- line 206, do not forget to put plus and minus symbol por DO, pH and EC;

Corrected

- Table 1, only one collum is sufficient to present diet composition since the only variation factor is the olive leaf extract;

In this table some other parameters are slightly different in different diets that is why we explained the diet composition in each group.

- Figure 1, random instead of "randum" replicate instead of "replica" hydrophila instead of "hidrophia";

Corrected

- line 235, explain how do you estimated LD50-96;

This dose was adopted from a previous research paper in the same field (48-           Abdel-Tawwab, M., El-Araby, D.A. (2021). Immune and antioxidative effects of dietary licorice (Glycyrrhiza glabra L.) on performance of Nile tilapia, Oreochromis niloticus (L.) and its susceptibility to Aeromonas hydrophila infection, Aquaculture, Volume 530,735828, https://doi.org/10.1016/j.aquaculture.2020.735828 )

- line 284, may the first full stop in the sentence is not corrected;

- Figure 2, Figure 3 Figure 4, please say in the legend what are the cases named with capital letters (A; B, C, D,...)

Corrected

- Line 510, A. hydrophila

- Line 514, take out the text filler;

- Figure 5, say in the legend what the cases E, F, G, H means;

Done

- Figure 6, say in the legend what the cases F, G, H means;

Done

- At the end of the discussion I would like to see what do you suggest for further research. For instance, do you think it is feasible to isolate and test only the considered positive substances in carp diet?

Our findings indicated that OLE, when used in C. carpio diets at a level of 0.1%, has a high capacity for growth performance by boosting nutrient utilization, whereas higher doses of the OLE must be avoided since they may induce oxidative stress.

Reviewer 3 Report

Please see in the attached report file.

GENERAL conclusion on the manuscript

This study is dealing with supplementation of carp diet with olive leaf extract, its characterization and feeding to fish in different doses. Throughout an extended physiological, histopathological and immunological investigation the effect of the olive leaf extract is presented and discussed. Overall, the manuscript is interesting, the experiment is well designed, the methods are adequate and well developed, the findings discussed deeply and compared with relevant studies.

However, there are some key problems in structure of the manuscripts that needs further improvement. Firstly, my suggestion is to move the chemical analysis, methods and related results of olive leave extract into Supplement part. These chapters (Task 2.3, Task 3.1, Table 4) contain information that undoubtedly are not strictly needed in understanding the results of the nutritional trial.

SPECIFIC comments:

Abstract

LINE 42. there are not mentioned 2nd, 3rd, 4th group in the text, please delete.

LINE 52 please check the downregulation statement of immune related genes. On Figure 9A there are upregulation compared to control. Modify accordingly.

Introduction

LINE114 Use more recent data on the carp production.

Materials and methods

LINE 160 include: yellow-brown extract (OLE)

TASK 3.2 In this section please divide the 2 techniques that was used for quality analysis and for quantification of most important components. Consequently, it is required to include separately all chemicals, columns, chromatographs and detectors as well. There are given several types of equipment: HPLC MS/MS, LC-ESI-MS, in Table 4: UPLC-ESI-MS/MS, in LINE 373 UPLC-MS/MS. Please define properly the equipment.

LINE 192 Please define TLC spots.

LINE 207 Stop the water quality from declining during the trial to maintain the water quality

LINE 220 Table 1. the olive leaf extract was given g/kg or in g/100 (%)? Please correct the definition according to the given amount.

Sum of the ingredients must be 100% not 99%, as is presented in the Table 1.

In this task details connected to the feed preparation have to be mentioned. The diet was accomplished through any extrusion process? If was made in laboratory condition, please indicate the equipment. The composition of the diets was calculated or measured?

Figure 1 I would suggest to change day 5dpc to day 65

LINE 375 Figure 2D not 2C

Figure 2 please change Ug/mL to µg/mL.

LINE 471 definition of the treatments is again wrong:0.1, 0.2, 0.3 g/kg %

LINE 486 figure B, definition is missing

Table 5 is missing number of fish n=15/treatment? Similarly, all the tables have to show the number of fish.

LINE 674 Figure 4 Figure 3B

LINE 683 marked decrease compared to control?  Something is not clear.

LINE 694 similar question: down-regulated? Not up-regulated? Please explain it.

Author Response

Dear respected reviewer, we thank you for your time and valuable comments and appreciate your effort to improve our manuscript. Here we mentioned point-by-point response to the reviewers’ comments in red colour and highlighted in the manuscript with yellow colour.

GENERAL conclusion on the manuscript

This study is dealing with supplementation of carp diet with olive leaf extract, its characterization and feeding to fish in different doses. Throughout an extended physiological, histopathological and immunological investigation the effect of the olive leaf extract is presented and discussed. Overall, the manuscript is interesting, the experiment is well designed, the methods are adequate and well developed, the findings discussed deeply and compared with relevant studies.

However, there are some key problems in structure of the manuscripts that needs further improvement. Firstly, my suggestion is to move the chemical analysis, methods and related results of olive leave extract into Supplement part. These chapters (Task 2.3, Task 3.1, Table 4) contain information that undoubtedly are not strictly needed in understanding the results of the nutritional trial.

This part is crucial to our work and is very important to the readers to be included in the main manuscript rather than in the supplementary file as it is the first time to perform such kind of analysis to the OLE.

SPECIFIC comments:

Abstract

LINE 42. there are not mentioned 2nd, 3rd, 4th group in the text, please delete.

Corrected

LINE 52 please check the downregulation statement of immune related genes. On Figure 9A there are upregulation compared to control. Modify accordingly.

These genes are pro-inflamatory and apoptotic genes that should be down-regulated by the substance which play a role as immune stimulant.

Introduction

LINE114 Use more recent data on the carp production.

These are the most recent statistics about carp production. These statistics are nor published every year.

Materials and methods

LINE 160 include: yellow-brown extract (OLE)

The extract is not strict yellow colour, it is yellowish-brown

TASK 3.2 In this section please divide the 2 techniques that was used for quality analysis and for quantification of most important components. Consequently, it is required to include separately all chemicals, columns, chromatographs and detectors as well. There are given several types of equipment: HPLC MS/MS, LC-ESI-MS, in Table 4: UPLC-ESI-MS/MS, in LINE 373 UPLC-MS/MS. Please define properly the equipment.

All these equipments were used for analysis of the components. And the techniques are separated as: analysis, characterisation and purification.

LINE 192 Please define TLC spots.

Done

LINE 207 Stop the water quality from declining during the trial to maintain the water quality

Done

LINE 220 Table 1. the olive leaf extract was given g/kg or in g/100 (%)? Please correct the definition according to the given amount.

Corrected

Sum of the ingredients must be 100% not 99%, as is presented in the Table 1.

Corrected

In this task details connected to the feed preparation have to be mentioned. The diet was accomplished through any extrusion process? If was made in laboratory condition, please indicate the equipment. The composition of the diets was calculated or measured?

The diet was commercially prepared and we added the OLE and then run the analysis for the different diets.

Figure 1 I would suggest to change day 5dpc to day 65

The time point by days is not of interest but the days post challenge are crucial in case of immune parameters and survival that is why we used days post challenge instead of the number of days from the start of the experiment.

LINE 375 Figure 2D not 2C

Corrected

Figure 2 please change Ug/mL to µg/mL.

Corrected

LINE 471 definition of the treatments is again wrong:0.1, 0.2, 0.3 g/kg %

Corrected

LINE 486 figure B, definition is missing

Table 5 is missing number of fish n=15/treatment? Similarly, all the tables have to show the number of fish.

Done

LINE 674 Figure 4 Figure 3B

Corrected

LINE 683 marked decrease compared to control?  Something is not clear.

This is because OLE is not growth promoting agent but with increasing the dose and as a compensatory mechanism the level of igf1B increase.

LINE 694 similar question: down-regulated? Not up-regulated? Please explain it.

Again these genes are pro-inflamatory and apoptotic genes that should be reduced or down-regulated when using an immune stimulant substance like OLE. Which indicate the immune modulatory effect of OLE.